# Self-bridging metamaterials surpassing the theoretical limit of Poisson's ratios

Jinhao Zhang[1], Mi Xiao [1] ✉, Liang Gao [1] ✉, Andrea Alù [2] & Fengwen Wang[3]

A hallmark of mechanical metamaterials has been the realization of negative Poisson's ratios, associated with auxeticity. However, natural and engineered Poisson's ratios obey fundamental bounds determined by stability, linearity and thermodynamics. Overcoming these limits may substantially extend the range of Poisson's ratios realizable in mechanical systems, of great interest for medical stents and soft robots. Here, we demonstrate freeform self-bridging metamaterials that synthesize multi-mode microscale levers, realizing Poisson's ratios surpassing the values allowed by thermodynamics in linear materials. Bridging slits between microstructures via self-contacts yields multiple rotation behaviors of microscale levers, which break the symmetry and invariance of the constitutive tensors under different load scenarios, enabling inaccessible deformation patterns. Based on these features, we unveil a bulk mode that breaks static reciprocity, providing an explicit and programmable way to manipulate the non-reciprocal transmission of displacement fields in static mechanics. Besides non-reciprocal Poisson's ratios, we also realize ultra-large and step-like values, which make metamaterials exhibit orthogonally bidirectional displacement amplification, and expansion under both tension and compression, respectively.

Microstructured metamaterials have been enabling the realization of exotic mechanical responses and deformation functionalities, including negative stiffness[1], negative compressibility[2,3], shape morphing[4–6], and twist modes[7]. In terms of the fundamental metric determining mechanical deformations, Poisson's ratio, metamaterials have been realized to support negative values based on auxetic deformation patterns, via bendable or buckled ligaments[8–12], or rotatable nodes[13–15]. The range of admissible Poisson's ratios for isotropic materials is [−1, 0.5]. Anisotropy may expand this range, but based on the orthotropic constitutive law, thermodynamics predicts a general bound on Poisson's ratios in the linear and stable elastic regime[16], namely $0 \leq v_{ij} v_{ji} < 1$ (Fig. 1f), where $v_{ij}$ and $v_{ji}$ are Poisson's ratios in two orthogonal directions ($i = 1, 2, 3$; $j = 1, 2, 3$; $i \neq j$) and micropolar elasticity is not considered[17]. This bound includes positive Poisson's ratios (Fig. 1a) in ordinary materials and negative Poisson's ratios (Fig. 1b) in auxetic metamaterials, but there is a large range of

unexplored space out of this bound (Fig. 1f). Designing Poisson's ratios surpassing the thermodynamic limit may realize extraordinary deformation patterns and substantially extend the functional applications of modern mechanical devices, such as soft robots[18] and biomedical equipment[19]. For instance, the Poisson's ratios $v_{12} v_{21} < 0$ (Fig. 1c) break the symmetry of constitutive tensors that originates from Maxwell–Betti reciprocity theorem[20,21], which predicts non-reciprocal transmission of the displacement field (Supplementary Note 1). Reciprocity stems from microscopic reversibility and time-reversal symmetry, which is a fundamental property in many physical systems, including electromagnetic[22], acoustic[23], elastodynamics[24], and elasticity[25]. Breaking static reciprocity associated with Poisson's ratios surpassing the thermodynamic limit may offer brand new opportunities to enrich the functionalities of mechanical systems, in the same way as acoustic nonreciprocity has enabled the creation of modern acoustic devices, including one-way mirrors and

[1]State Key Laboratory of Digital Manufacturing Equipment and Technology, Huazhong University of Science and Technology, 430074 Wuhan, China. [2]Photonics Initiative, Advanced Science Research Center, City University of New York, New York, NY 10031, USA. [3]Department of Civil and Mechanical Engineering, Technical University of Denmark, Koppels Allé, Building 404, 2800 Kongens Lyngby, Denmark. ✉e-mail: xiaomi@hust.edu.cn; gaoliang@mail.hust.edu.cn

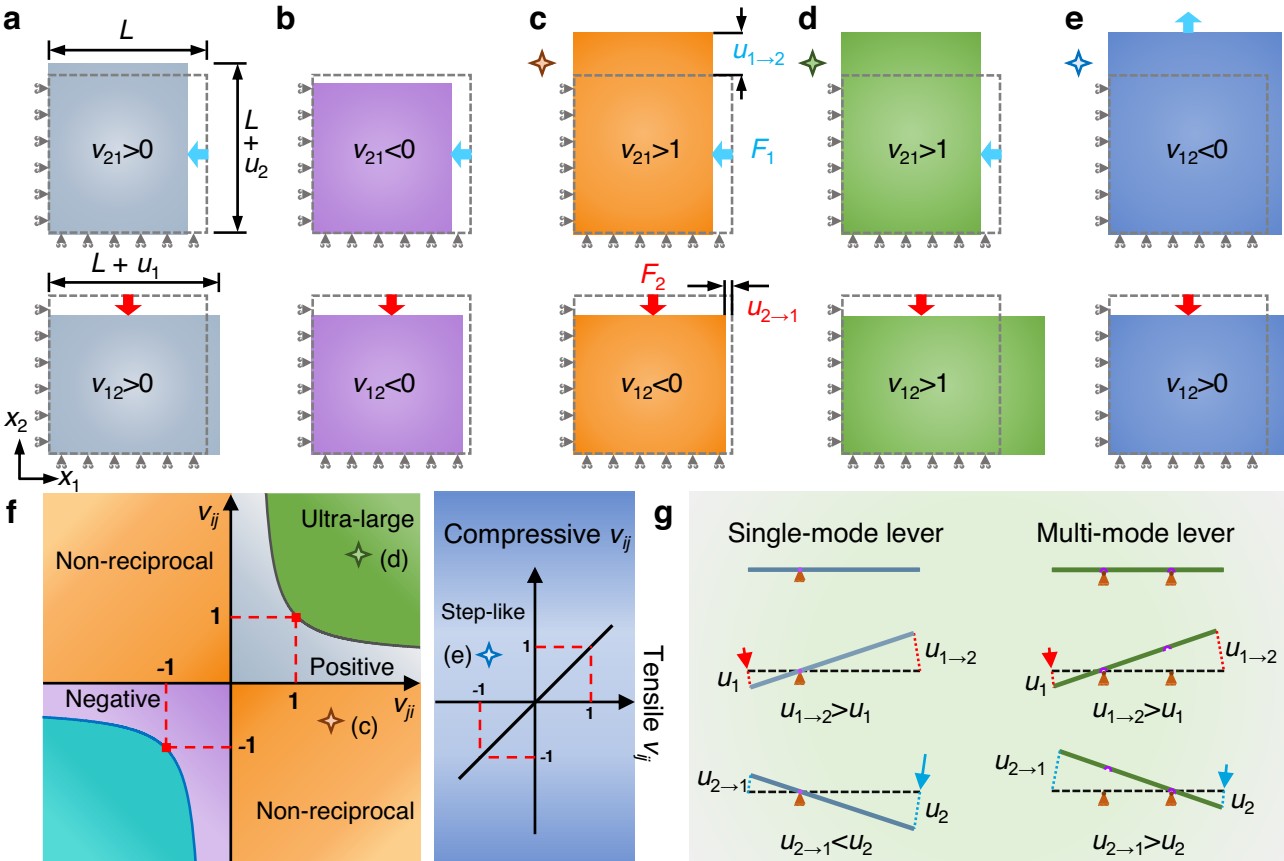

**Fig. 1 | Deformation patterns associated with different Poisson's ratios and tailored microstructures. a, b** Traditional deformation patterns of positive and negative Poisson's ratios. The gray dashed boxes indicate undeformed structures of length $L$. The blue and red arrows represent the load directions, and $u_1$ and $u_2$ are displacements in the $x_1$ and $x_2$ directions, respectively. **c–e** Unusual deformation patterns associated with non-reciprocal, ultra-large, and step-like Poisson's ratios, respectively (Supplementary Notes 1–3). $u_{2\to1}$ ($u_{1\to2}$) is the output displacement in the $x_1$ ($x_2$) direction under the load $F_2$ ($F_1$). **f** Comparison between ultra-large, non-reciprocal Poisson's ratios and the thermodynamic limit $0 \leq v_{ij}v_{ji} < 1$, and comparison between step-like Poisson's ratios and invariant $v_{ij}$ under compression or tension in the linear elastic regime. **g** Displacement amplification of two different levers. The single-mode lever with one fulcrum can only amplify the input displacement on the left, namely, $u_{1\to2} > u_1$ and $u_{2\to1} < u_2$. The multi-mode lever with two detachable fulcrums can amplify the input displacements on both the left and right by changing different fulcrums, namely, $u_{1\to2} > u_1$ and $u_{2\to1} > u_2$.

circulators[23,26]. The thermodynamic limit is derived from the linear elastic regime with the assumption of infinitesimally small strain, yet the introduction of nonlinearities has so far failed to produce Poisson's ratios surpassing the limit[27,28].

Like a single-mode lever with one fixed fulcrum (Fig. 1g), the thermodynamic limit ($0 \leq v_{ij}v_{ji} < 1$) only allows displacement amplification in one direction with a Poisson's ratio larger than 1. However, a multi-mode lever with two detachable fulcrums can overcome this limit and realize bidirectional displacement amplification by changing rotation modes via different fulcrums (Fig. 1g). In terms of internal connectivity, the lever system is reconfigured due to the transition between the connection and separation states of the fulcrums and the lever, and then the lever shows different rotation modes.

In this work, inspired by the opportunities stemming from the topological reconfiguration of detachable fulcrums, we use predefined slits in the continuum to mimic detachable fulcrums. We develop a powerful inverse design framework that combines predefined slits and topology optimization, realizing self-bridging mechanical metamaterials that synthesize multi-mode microscale levers and exhibit Poisson's ratios surpassing the thermodynamic limit. In terms of internal connectivity, the designed mechanical metamaterials respond to different loads with changed topological configurations induced by the self-bridging of predefined slits via self-contacts. The rotation behaviors of microscale levers in the metamaterials are then changed, and effective constitutive tensors of the metamaterials no longer obey

the symmetry and invariance under different load scenarios. We can therefore achieve self-bridging metamaterials with Poisson's ratios surpassing the conventional limits, enabling inaccessible deformation patterns, including one-way displacement amplification with broken static reciprocity (non-reciprocal Poisson's ratios, $v_{ij}v_{ji} < 0$, in Fig. 1c, f), orthogonally bidirectional displacement amplification (ultra-large Poisson's ratios, $v_{ij}v_{ji} > 1$, in Fig. 1d, f), and transverse expansion under both longitudinal tension and compression (step-like Poisson's ratios, compressive $v_{ij} > 0$ and tensile $v_{ij} < 0$, in Fig. 1e, f). Particularly, the nonlinearity of self-contacts that is stronger than geometric and material nonlinearities can be active under small strains by setting zero initial distance between self-contact surfaces. Thus, the aforementioned deformation patterns can be activated by small strains and maintained under finite strains.

## Results
### Design, simulations, and experiments
We developed an inverse design framework to generate the freeform microstructural configurations of self-bridging mechanical metamaterials with target Poisson's ratios surpassing the thermodynamic limit (see Methods section "Design method of self-bridging metamaterials"). In this framework, the design problem is described by the optimization problem in Eq. (7), where the target Poisson's ratios are realized by constraining the error between actual and prescribed values. Slits are predefined in the unit cell (Fig. 2a, b), and then the

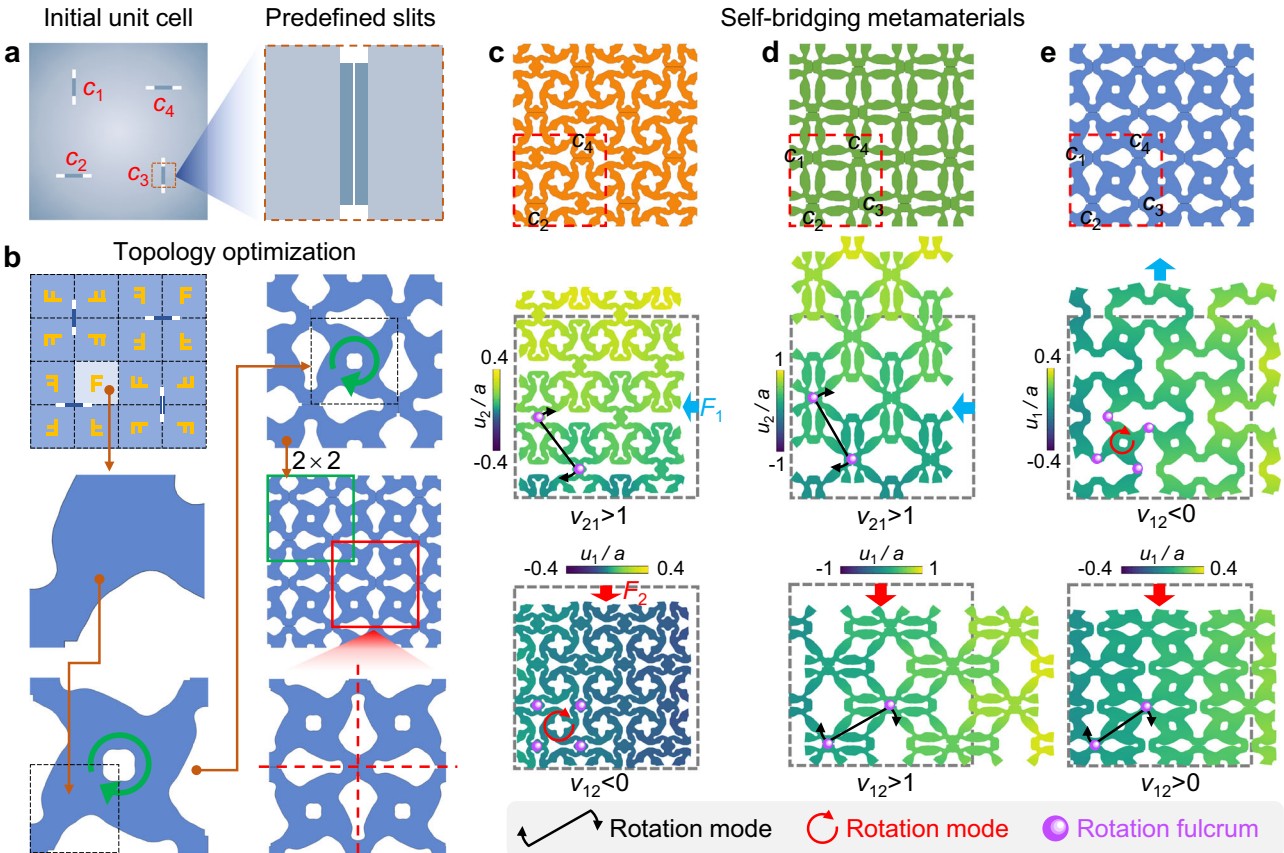

**Fig. 2 | Design method, simulation, and comparison. a** Initial unit cell with pre-defined slits ($c_1$, $c_2$, $c_3$, and $c_4$). The locally enlarged view of $c_3$ is shown on the right. **b** Topology optimization and symmetry. **c–e** Designed microstructures and simulations for non-reciprocal, ultra-large, and step-like Poisson's ratios, respectively.

Each microstructure consists of $2 \times 2$ unit cells with lattice constant $a$, one of which is boxed by red dashed lines. Prescribed slits are labeled $c_1$, $c_2$, $c_3$, and $c_4$. Each color bar shows the output displacement field in the $x_1$ or $x_2$ direction. The rotation modes and fulcrums are denoted by arrows and balls, respectively.

topological configuration of each mechanical metamaterial is independently designed by setting target Poisson's ratios in the optimization formulation. In the developed approach, elemental density variables are introduced to describe the material distribution[29–35]. The independently designed region in the approach is indicated in gray, and orange "F" is used to indicate the symmetry in the cell (Fig. 2b). The cell has four-fold rotational symmetry in the entire domain, and mirror symmetry exists in its quarter at each corner. The slits are distributed along the axis of mirror symmetry. Two slits ($c_2$ and $c_4$) are predefined for the design of non-reciprocal Poisson's ratios, while four slits ($c_1$, $c_2$, $c_3$, and $c_4$) are predefined for the design of ultra-large and step-like Poisson's ratios. The geometrical parameters of the slits are provided in Supplementary Fig. 1. Figure 2c–e show the designed microstructures with target Poisson's ratios $v_{21}^* = 1.4$ and $v_{12}^* = -0.5$, $v_{21}^* = v_{12}^* = 3$, and tensile $v_{12}^* = -0.8$ and compressive $v_{12}^* = 0.8$, respectively. The locally enlarged views of the slits in the designed metamaterials are shown in Supplementary Fig. 1. The distributions of self-contacts induced by predefined slits and the topology of the microstructures are significantly different from those in other design frameworks using self-contacts[6,11], which are built for different properties. Numerical simulations with $2 \times 2$ unit cells (Fig. 2c–e) were performed to evaluate Poisson's ratios using finite-element models with periodic boundary conditions (Supplementary Fig. 2) and zero initial distance between self-contact surfaces in COMSOL Multiphysics 6.0. In this work, periodic boundary conditions are constructed based on orthotropy. More general periodic boundary conditions can be found in the studies[36–38]. We arranged the designed unit cells periodically and fabricated hyperelastic samples by molding (see Methods section "Measurement"). Poisson's ratios of these

samples are evaluated by uniaxial compression and tension tests (Supplementary Fig. 3).

## Non-reciprocal Poisson's ratios

The metamaterial in Fig. 2c expands and contracts under transverse and longitudinal compressions (strains of −10%), respectively, which results in positive and negative Poisson's ratios (Fig. 3a), respectively. In the numerical simulations, all microstructural slits open under transverse compression. However, when the microstructure is longitudinally compressed, the self-bridging of slits at $c_2$ and $c_4$ reconfigures the microstructural topology with a change in internal connectivity. Then, the microscale levers show two different rotation modes: (1) for transverse compression, the microstructure rotates about two fulcrums; (2) for longitudinal compression, the arms at $c_2$ and $c_4$ drive the rotation of the microstructure. These two different rotation behaviors result in positive and negative Poisson's ratios, $v_{21} > 0$ and $v_{12} < 0$, respectively (Fig. 3a). We found the experimental deformation patterns (Fig. 4a) and Poisson's ratios (Fig. 3a) agree well with the simulated ones. These Poisson's ratios violate the lower bound of the thermodynamic limit ($v_{ij}v_{ji} \geq 0$) that is derived from static reciprocity (Supplementary Note 1). With positive elastic moduli ($E_1 > 0$ and $E_2 > 0$ in Fig. 3b), $E_2 v_{21} > 0 > E_1 v_{12}$ (Fig. 3c), which demonstrates that the designed metamaterial breaks the symmetry of its constitutive tensors ($E_j v_{ij} = E_i v_{ji}$), stemming from static reciprocity. Hence, we can predict the emergence of a static non-reciprocal bulk mode ($F_1 u_{2 \to 1} > 0 > F_2 u_{1 \to 2}$), different from the one in "fishbone" metamaterials designed by breaking geometrical symmetry and introducing geometrical nonlinearity[25]. The Poisson's ratios $v_{21} > 1$ and $0 > v_{12} > -1$ (Fig. 3a) predict that the metamaterial can amplify the input displacement field unidirectionally. Owing to the tiny slits and

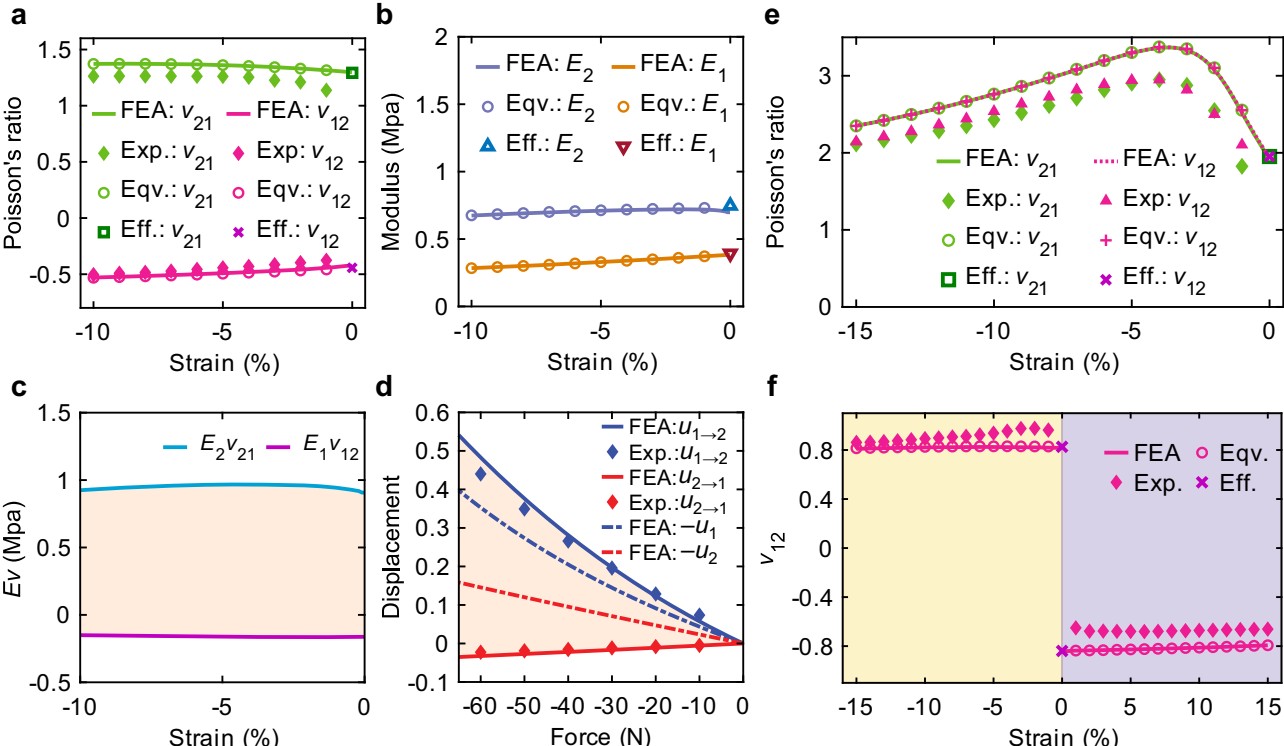

**Fig. 3 | Poisson's ratio, elastic moduli, and non-reciprocity evaluations. a** Non-reciprocal Poisson's ratio. "FEA (finite-element analysis)," "Exp. (experiment)," "Eqv. (equivalent)," and "Eff. (effective)" correspond to simulations of designed unit cells, experiments about designed unit cells, simulations of equivalent models, and effective material parameters, respectively. **b** Elastic moduli $E_1$ and $E_2$ of the metamaterial with non-reciprocal Poisson's ratios in two orthogonal directions. **c** Products of elastic moduli and Poisson's ratios. The shaded area corresponds to the difference between $E_2 v_{21}$ and $E_1 v_{12}$. **d** Output displacement $u_{1\to2}$ ($u_{2\to1}$) and input displacement $u_1$ ($u_2$) for different values of $F_1$ ($F_2$). The shaded area corresponds to the difference between $u_{1\to2}$ and $u_{2\to1}$. **e** Ultra-large Poisson's ratios. **f** Step-like Poisson's ratios. The two shaded areas correspond to compression and tension strains, respectively.

strong nonlinearity of self-contacts, a huge difference between $E_2 v_{21}$ and $E_1 v_{12}$ emerges under small strains, as shown in Fig. 3c. Hence, small strains may activate the non-reciprocal transmission of displacement fields in the designed metamaterial.

To interpret the relation between the self-bridging feature and broken static reciprocity, we built two equivalent models in Supplementary Fig. 4a, b, where the slits closed under compression are replaced by solid connections. Each equivalent model is only valid for a special load case indicated by a gray arrow, which is determined by the mimicked metamaterial under the same load case (Supplementary Fig. 4). The internal connectivities of the equivalent models are different owing to the additional solid connections. The equivalent models were numerically simulated without defining self-contacts. The Poisson's ratios and elastic moduli of the equivalent models are almost the same as those of the designed metamaterial (Fig. 3a, b). Thus, the designed metamaterial (Fig. 2c) can be substituted by these equivalent models. To explicitly distinguish these equivalent models, we calculated the effective compliance matrix of each equivalent model using the representative volume element method (Supplementary Note 4)[39] based on the linear orthotropic constitutive law. The evaluated effective compliance matrices of our equivalent model are significantly different (Supplementary Equations (13, 14)), demonstrating topological reconfiguration of the metamaterial induced by self-bridging slits. Thus, the self-bridging feature of the metamaterial enables the variation of constitutive tensors under different load scenarios, which breaks the symmetry $E_j v_{ji} = E_i v_{ij}$ in static reciprocity and enables surpassing the lower bound ($v_{ij} v_{ji} \geq 0$). In mechanical systems under small strains, general geometric and material nonlinearities may vanish[25], but self-contacts can occur via zero initial distance between self-contact surfaces. The effective Poisson's ratios and elastic moduli of the equivalent models agree well with those of the designed metamaterial

under small strains (Fig. 3a, b), which demonstrates that the designed self-bridging metamaterial can break the static reciprocity under small strains.

The broken static reciprocity associated with Poisson's ratios was tested in the designed metamaterial (Fig. 4d). Figure 4d shows the experimentally observed non-reciprocal deformation patterns of the metamaterial, which was supported by rollers to create sliding boundary conditions, and a force $F_1 = -60$ N ($F_2 = -60$ N) was applied at its right (top) side. Under the same force, $u_{1\to2} > 0 > u_{2\to1}$, and $|u_{1\to2}|$ is an order of magnitude greater than $|u_{2\to1}|$, as shown in Fig. 4d, which is evidence of non-reciprocal transmission of the displacement fields. In addition, the output displacement $|u_{1\to2}|$ is large than the input displacement $|u_1|$, while the output displacement $|u_{2\to1}|$ is smaller than the input displacement $|u_2|$. Thus, there is a unidirectionally amplified non-reciprocal displacement field, and the amplification performance well fits the prediction of designed Poisson's ratios. Poisson's ratio is a fundamental property that directly quantifies the relation between input ($u_i$) and output ($u_{i\to j}$) displacement. Hence, our designed self-bridging metamaterial may provide an explicit and programmable way to manipulate non-reciprocal transmission of the displacement field via designing Poisson's ratios.

### Ultra-large Poisson's ratios
The metamaterial in Fig. 2d shows large expansibility under compressions (strains of −15%) in both orthogonal directions. The output displacement is much larger than the input, which results in ultra-large Poisson's ratios (with a maximum value of approximately 3) in both orthogonal directions (Fig. 3e). The product of Poisson's ratios in two orthogonal directions is much larger than 1, namely $v_{12} v_{21} > 1$. We found the experimental deformation patterns (Fig. 4b) and Poisson's ratios (Fig. 3e) agree well with the simulated ones. Hence, this metamaterial

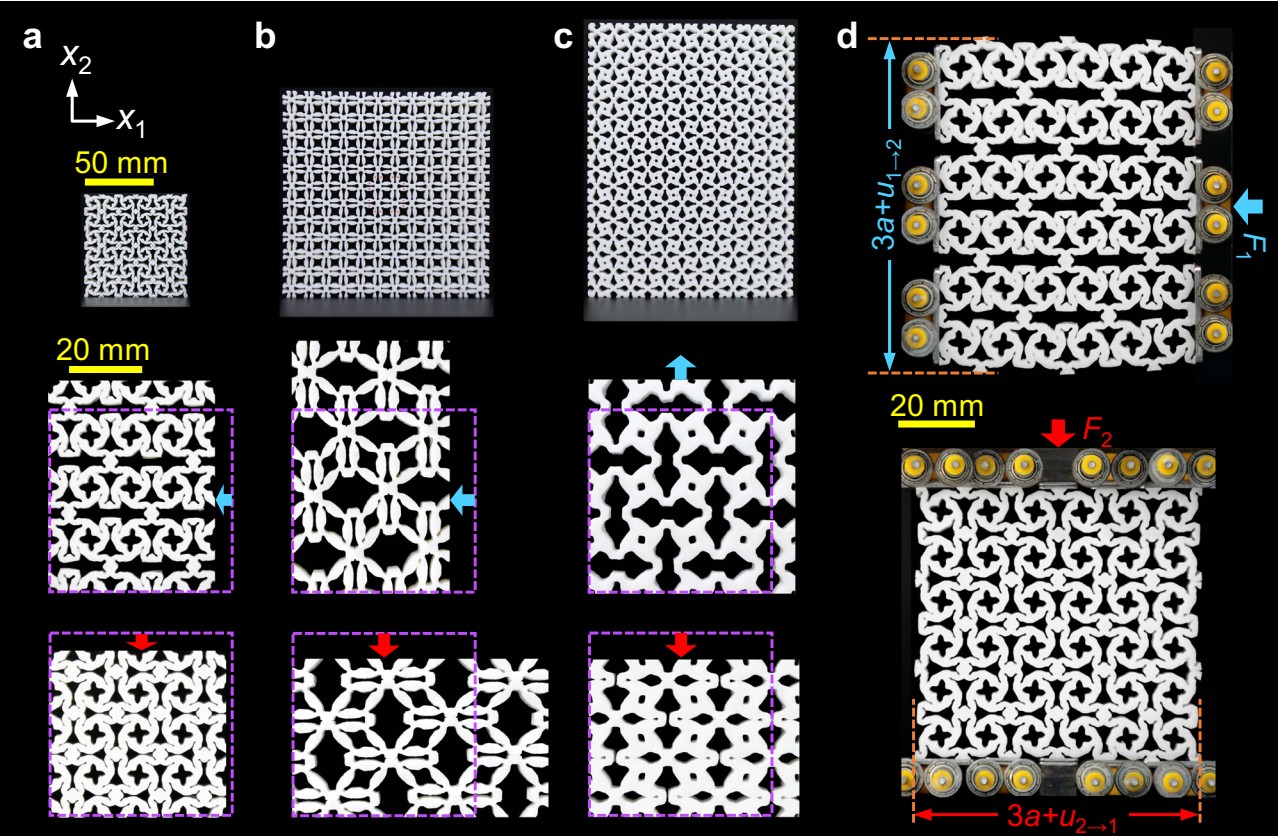

**Fig. 4 | Experiments. a–c** Samples of designed metamaterials and snapshots of the deformation patterns for non-reciprocal (Supplementary Video 1), ultra-large (Supplementary Video 2), and step-like (Supplementary Video 3) Poisson's ratios, respectively. **d** Experiments about non-reciprocity under transverse and longitudinal loads. The structure is supported by rollers that are compressed using an aluminum alloy plate. In Fig. **a**–**c**, undeformed samples have the same scale bar of 50 mm, and the deformed samples have the same scale bar of 20 mm. In Fig. 4d, the scale bar of 20 mm applies to all structures.

achieves stable and ultra-large Poisson's ratios surpassing the thermodynamic limit ($v_{ij}v_{ji} < 1$) (Supplementary Note 2). In the numerical simulations, the locations of rotation fulcrums (self-contacts) under transverse compression are $c_1$ and $c_3$, while the locations change to $c_2$ and $c_4$ under longitudinal compression (Fig. 2d). In terms of internal connectivity, different locations of self-contacts make the microstructure exhibit different topological configuration. The length of microscale levers between two fulcrums increases under compression, which amplifies the input displacement, and then the changed rotation fulcrums lead to amplification in both orthogonal directions. The bidirectional displacement amplification mechanism of the metamaterial is similar to a multi-mode lever with two detachable fulcrums, where input displacements on both the left and right can be amplified by changing the fulcrum (Fig. 1g).

The observed ultra-large Poisson's ratios are analyzed by effective material parameters. We built two equivalent models (Supplementary Fig. 4c, d) to mimic the metamaterial (Fig. 2d) under different load scenarios, which have different internal connectivities owing to the additional solid connections. The Poisson's ratios of the equivalent models are almost the same as those of the designed metamaterial (Fig. 3e). Thus, the designed metamaterial can be well substituted by these equivalent models. The upper bound $v_{ij}v_{ji} < 1$ is deduced by assuming an invariant positive-definite matrix to ensure a positive strain energy density. However, in the effective compliance matrices of these two equivalent models (Supplementary Equations (15, 16)), the values of the first and second elements on the principal diagonal are swapped, which demonstrates that the microstructural topology of the metamaterial is reconfigured as the locations of the self-bridging slits change. The constitutive tensors of the designed metamaterial vary under different load cases, and thus the upper bound can be violated by the self-bridging metamaterial.

## Step-like Poisson's ratios

The metamaterial in Fig. 2e expands transversely under both longitudinal tension (a strain of 15%) and compression (a strain of −15%), which results in step-like Poisson's ratios (compressive $v_{12} > 0$ and tensile $v_{12} < 0$, Fig. 3f). Such Poisson's ratios are inaccessible for materials in the linear elastic regime owing to the assumption of invariance of the constitutive tensors (Supplementary Note 3). The metamaterial is designed in the continuum instead of a discrete rigid body[15], and it realizes the expansibility under both tension and compression, different from the contractility induced by bending beams[40,41]. In the numerical simulation under compression, in terms of internal connectivity, the microstructural topology is reconfigured by self-bridging slits at $c_2$ and $c_4$ in the microstructure (Fig. 2e). Then, the microscale levers show different rotation modes: (1) Under tension, all slits are pulled apart, and the microstructure rotates with anti-chirality; (2) Under compression, $c_1$ and $c_3$ open while $c_2$ and $c_4$ close, and the microstructure rotates about fulcrums at $c_2$ and $c_4$. Only the first rotation mode activates the auxeticity, and thus the metamaterial exhibits step-like Poisson's ratios. We found the simulated deformation patterns agree well with the experimental ones (Fig. 4c).

We built two equivalent models (Supplementary Fig. 4e, f) to mimic the metamaterial (Fig. 2e) under different load scenarios. The Poisson's ratios of the equivalent models are almost the same as those of the designed metamaterial (Fig. 3f). Thus, these equivalent models can accurately describe the metamaterial. The calculated effective compliance matrices of these two equivalent models are significantly

different (Supplementary Equations (17, 18)). Because the constitutive tensors of the designed metamaterial are no longer invariant under different loads, the self-bridging metamaterial can show different Poisson ratios under different loads.

### Customization

These unusual Poisson ratios can be tailored by adjusting the target values in the inverse design framework. More designed metamaterials and corresponding deformation patterns are shown in Supplementary Fig. 5. These metamaterials are designed based on the target non-reciprocal Poisson's ratios $v_{21}^* = 1.8$ and $v_{12}^* = -0.5$, ultra-large Poisson's ratios $v_{21}^* = v_{12}^* = 2$, and step-like Poisson's ratios $v_{12}^* = -0.4$ under tension and $v_{12}^* = 0.4$ under compression (Supplementary Fig. 6), respectively. This shows that the realized deformation patterns can be customized by designing purely passive microstructures.

## Discussion

Mechanical metamaterials surpassing the thermodynamic limit of Poisson's ratios were designed and realized through inverse design based on predefined slits for self-contacts and topology optimization. Symmetry and invariance of constitutive tensors are basic assumptions for the theoretical limit of Poisson's ratios in the linear elastic regime. In terms of internal connectivity, the microstructural topologies of the designed mechanical metamaterials are reconfigured by the self-bridging slits, driving different rotation behaviors of microscale levers. Then, under different loads, the constitutive tensors of the self-bridging metamaterials no longer obey symmetry and invariance, enabling Poisson's ratios that surpass the conventional limits.

These metamaterials possess inaccessible deformation patterns, including one-way displacement amplification with broken static reciprocity, orthogonally bidirectional displacement amplification, and transverse expansion under both longitudinal tension and compression. These deformation patterns are active under both large and small strains by setting zero initial distance between self-contact surfaces, and can be tailored by the topological design of purely passive microstructure. Based on non-reciprocal Poisson's ratios, we demonstrate the emergence of a bulk mode that breaks static reciprocity, which opens new avenues to manipulate non-reciprocal transmission of the displacement field via designing Poisson's ratios. Our work may substantially extend the applications of Poisson's ratios in modern devices, for example, mechanical energy harvesters[42] with oneway amplified displacement field, strain sensors[43] with bidirectional ultra-large displacement amplification, civil protection equipment[44], and biomedical stents[45] with expansibility of tension and compression, and soft robots[18] with flexible structural deformation.

## Methods

### Design method of self-bridging metamaterials

The strain energy density $W_M$ of the two-term Mooney–Rivlin model is expressed as

$$W_M = A_{10}(I_1 - 3) + A_{01}(I_2 - 3) \tag{1}$$

where $I_1$ and $I_2$ are the first and second invariants of the right Cauchy–Green deformation tensor, respectively. $A_{10}$ and $A_{01}$ can be determined by fitting the relation between engineering strain and stress in uniaxial tension tests of the base material (Supplementary Fig. 7).

A design method with density-based topology optimization is built to design the unit cell, in which a set of element-wise design variables, $\xi \in [0, 1]$, is introduced to describe the material distribution. The unit cell is constrained by periodic boundary conditions (Supplementary Fig. 2). To avoid a checkerboard pattern and enhance boundary discreteness, the design variables are first filtered with the

Helmholtz filter[24] and then projected using the tanh function[25] via

$$-\left(\frac{r_f}{2\sqrt{3}}\right)^2 \nabla \tilde{\xi}(\mathbf{r}) + \tilde{\xi}(\mathbf{r}) = \xi(\mathbf{r}), r_f > 0, \mathbf{r} = \{x_1, x_2\}, \mathbf{r} \in \Omega_D \tag{2}$$

$$\bar{\xi} = \frac{\tanh(\beta\eta) + \tanh\left(\beta(\tilde{\xi} - \eta)\right)}{\tanh(\beta\eta) + \tanh(\beta(1 - \eta))} \tag{3}$$

where $r_f$ denotes the desired spatial filtering radius, $r_f = 0.05a$ in this work. $\xi$ is a continuous representation of the unfiltered design field. $\tilde{\xi}$ and $\bar{\xi}$ denote the filtered and projected design variables, respectively. $\beta$ increases with iteration to enhance boundary discreteness. SIMP interpolation[24] is used to establish the relation between the projected variables and material parameters:

$$A''_{10}(\bar{\xi}) = A'_{10} + \bar{\xi}^3(A_{10} - A'_{10}) \tag{4}$$

$$A''_{01}(\bar{\xi}) = A'_{01} + \bar{\xi}^3(A_{01} - A'_{01}) \tag{5}$$

where $A'_{10} = 10^{-9}A_{10}$ and $A'_{01} = 10^{-9}A_{01}$ Then, in Eq. (1), $A_{10}$ and $A_{01}$ are replaced by $A''_{10}$ and $A''_{01}$, respectively.

For accessible feature size in fabrication, a three-case robust formulation[25] is used in which eroded, normal, and dilated manufacturing processes are mimicked with $\eta = \eta_e$, $\eta_i$, and $\eta_d$, respectively. $\eta_e$, $\eta_i$, and $\eta_d$ are set to 0.55, 0.5, and 0.45, respectively.

To enhance numerical stability under large compression strain, we adopt the energy interpolation form[26]

$$W = W_M(\gamma \mathbf{u}) + (1 - \gamma^2)W_L(\mathbf{u}) \tag{6}$$

where $W_L$ is the strain energy density of the linear elastic model in small-deformation theory, $\mathbf{u}$ is the displacement field, and $\gamma$ is the interpolation factor.

The stiffness of the metamaterial is maximized using a formulated density-based topology optimization model

$$\begin{aligned} \max_{\bar{\xi}} \min_{\eta} \ & W(\bar{\xi}, \eta, \varepsilon_1^t) \\ s.t. \quad & \max_{\varepsilon_j^t}\left(\left(v\left(\bar{\xi}, \eta_i, \varepsilon_j^t\right) - v^*\left(\varepsilon_j^t\right)\right)^2\right) < \zeta, j = 1, \dots, n \\ & \frac{\mathbf{s}^\mathsf{T}\bar{\xi}(\xi, \eta_d)}{a^2} \leq s^* \\ & \mathbf{0} \leq \bar{\xi} \leq \mathbf{1} \end{aligned} \tag{7}$$

where $v$ and $v^*$ are the actual and prescribed properties for a given target strain $\varepsilon_j^t$, respectively; $n$ is the total number of target strains; $\zeta$ is used to relax the constraint on the Poisson's ratio, which is set to $0.05^2$. $a^2$ is the area of the unit cell; $\mathbf{s}$ is the elemental area vector; the prescribed area fraction is $s^* = 0.5$. The method of moving asymptotes[27] is the optimizer used to solve the model, where the gradient of the objective and constraints are calculated via the adjoint method[28]. The update processes of the microstructural topologies are presented in Supplementary Fig. 8.

### Sample fabrication

The samples with non-reciprocal, ultra-large, and step-like Poisson's ratios consisted of $3 \times 3$, $6 \times 6$, and $6 \times 8$ unit cells with lattice constant $a = 25$ mm, as shown in Fig. 4a–c, respectively. The lattice constant was chosen according to the manufacturing limit. The samples of the designed unit cells were fabricated by molding. Before molding, all slits were connected with a solid. Because of the complexity of the shapes of the designed unit cells, the designed unit cell prototypes for molding were fabricated with a light-curing 3D printer (iSLA660). The processed prototypes were covered with a silica gel solution in boxes, and the air in

the liquid silica gel was eliminated with vacuum equipment. The molds were obtained after prototypes were removed from the cured silica gel solution. Then, the molds were placed into a vacuum pouring machine (HZK1000) and injected with mixed polyurethane raw material (T0387). The molds were removed to obtain the polyurethane samples when the material was cured. In postprocessing, the samples were knifed to recover the slits. For the material, $A_{10}$ and $A_{01}$ of the two-term Mooney–Rivlin model were −0.17 and 2.5 MPa, respectively, which were obtained by least-squares fitting of the relation between engineering strain and stress in a uniaxial tension test (Supplementary Fig. 7).

### Measurement

Tension and compression tests were performed using a HANDPI machine with a 500 N loading cell. For non-reciprocal Poisson's ratios, the sample in Fig. 4a was tested via transverse and longitudinal uniaxial compression, where the sample was supported by rollers to create sliding boundary conditions (Supplementary Fig. 3a). The thickness of the sample in Fig. 4a was 25 mm, which was chosen to avoid out-of-plane buckling. For ultra-large Poisson's ratios, the sample in Fig. 4b was tested via transverse and longitudinal uniaxial compression. For step-like Poisson's ratios, the sample in Fig. 4c was tested via longitudinal uniaxial tension (Supplementary Fig. 3b) and compression (Supplementary Fig. 3c). The thickness of the samples in Fig. 4b, c was 10 mm. Under compressions, the samples in Fig. 4b, c were held vertically between two polymethyl methacrylate sheets, which were held 10.5 mm apart (Supplementary Fig. 3b). Under tension, the top and bottom 1×6 unit cells of the sample in Fig. 4c were fastened between two aluminum alloy plates (Supplementary Fig. 3c). We repeated all experiments five times for evaluating Poisson's ratios. The tests were monitored using a camera (EOS M6 Mark II) with 3840×2160 pixels. The videos were analyzed with MATLAB Image Processing Toolbox. We obtained Poisson's ratios for different strains by tracking the displacement vectors of red markers on the samples (Supplementary Fig. 3d). The Poisson's ratios were approximated as $\tilde{\nu}_{12} = \bar{u}_1/\bar{u}_2$ and $\tilde{\nu}_{21} = \bar{u}_2/\bar{u}_1$, where $\bar{u}_1$ ($\bar{u}_2$) is the average displacement difference between two columns of vertical red markers (two rows of horizontal markers).

### Data availability

The data of effective compliance matrixes are provided in Supplementary Note 4. All other data that support the plots within this paper and other findings of this study are available from the corresponding authors upon request.

### Code availability

The optimization algorithms necessary to perform the inverse design are described in the main text, Methods, and Supplementary Information. All other codes that support the plots within this paper and other findings of this study are available from the corresponding authors upon request.

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

## Acknowledgements
We thank Prof. Roderic Lakes at the University of Wisconsin and Prof. Ole Sigmund at the Technical University of Denmark for their helpful discussions. This work is supported by China National Postdoctoral Program for Innovative Talents (BX20220124, J.Z.), Project Funded by China Postdoctoral Science Foundation (2022M710055, J.Z.), National Key Research and Development Program of China (2020YFB1708300, M.X.), XPLORER PRIZE (L.G.), and Villum Investigator Project Innotop from the Villum Foundation (F.W.).

## Author contributions
J.Z., M.X., and F.W. conceived the research problem. J.Z. and F.W. developed the design method. J.Z., M.X., and L.G. fabricated the samples and conducted experiments. J.Z. performed the finite-element simulations and wrote the original draft. J.Z., M.X., L.G., A.A., and F.W. analyzed the data, interpreted the results, and edited the manuscript.

## Competing interests
The authors declare no competing interests.
