## [Peer Review File · Nature Communications]

Self-bridging metamaterials surpassing the theoretical limit of Poisson's ratiosREVIEWER COMMENTS

Reviewer #1 (Remarks to the Author):

The authors introduce three types of self-bridging metamaterials, which achieve non-trivial Poisson's ratios, respectively non-reciprocal Poisson's ratios, ultra-large Poisson's ratios and step-like Poisson's ratios. The three metamaterials are designed based on predefined slits using topology optimization. The results have been verified by experiments, simulations and equivalent models.

The main novelty of this article is presenting the concept of three non-trivial Poisson's ratios using self-bridging metamaterials. Many studies [1][2] have used self-contact (or self-bridging) to design mechanical metamaterials. The first property, non-reciprocal Poisson's ratio widely exist in anisotropy materials. It is not surprising that the anisotropy structure the authors presented shows non-reciprocal Poisson's ratio. For example, Fig.3(a) and (b) of the ref[3] clearly show non-reciprocal Poisson's ratio. The second property, ultra-large Poisson's ratios, has been studied in ref[4] and also shown in Fig.3(a) of the ref[3]. The third property, step-like Poisson's ratio also has been studied by several researchers, see Fig.1 of the ref [5]. I agree that although all concepts have been studied before, to the best of my knowledge, the combination of these concepts is new. But the fundamental rule or the design guide of the presented self-bridging metamaterials is unclear to me. The design is based on the topology optimization, it is unclear that which part of the structure is essential to achieve the properties that the authors presented.

Here more detailed comments and requests for clarification:

1. I like that the authors use a lever with two fulcrums (Fig.1 h) to explain how to achieve a bidirectional amplification. However, I did not fully understand how the authors apply this multiple-fulcrum lever mechanism to the three designs in Fig 2. c,d and e. I could see both Fig.1 h and Fig. 2d have fulcrums and both show a 'bidirectional' amplification, but I could not see a further connection.

2. Authors design the three metamaterials using topology optimization. The structure shown in Fig. 2c shows non-reciprocal Poisson's ratios, but it should also be able to show the step-like Poisson's ratios, i.e. compressive $\nu_{12} < 0$ and tensile $\nu_{12} > 0$. Why do the authors design another structure shown in Fig. 2e? Similarly, if there are predefined slits at position C1 and C3 as well in the structure shown in Fig. 2c, then it should also be able to achieve ultra-large Poisson's ratio, i.e. compressive $\nu_{12} > 1$, compressive $\nu_{12} > 1$.

So by only adjusting the predefined slits, all the three non-trivial Poisson's ratios can be realized based on the orange structure (Fig. 2c). But the authors design three different structures to show three non-trivial properties of Poisson's ratios. Do different designs add any new information? If it is just due to topology optimization, the novelty and influence will be greatly reduced.

3. So the main question would be what is the design guide? Which geometry parameters are essential to

achieve the non-trivial Poisson's ratios? By adding predefined slits, the symmetry of tension and compression is broken, so ν_{12} (compressive) \neq ν_{21} (tensile). So all the three structures should show 'step-like Poisson's ratios'. The green one and blue one have step-like Poisson's ratio along both horizontal and vertical directions. The orange one has a step-like Poisson's ratio only along vertical direction. The green one and blue one also have C4 rotation symmetry, which makes $\nu_{12}=\nu_{21}$. Therefore, if $\nu_{12}>1$, then $\nu_{21}>1$ as well, named 'ultra-large Poisson's ratios' in the article. While the orange one does not have C4 rotation symmetry, so $\nu_{12} \neq \nu_{21}$, named non-reciprocal Poisson's ratios in the article. On the top of the symmetries, if the chiral motion is triggered, then the Poisson's ratio is positive.

4. There is still some space to improve the structure of the manuscript. Some parts are redundant. For example, Fig.1 a-e show different combinations of Poisson's ratios, but Fig. 1f summarizes all in a concise way. Fig 1 i,j,k and the top row of Fig 2 c,d,e show the same design. Similarly, Fig.2 c,d,e and Fig 4 show the information as well. Fig.2 f and g doesn't add much information to this article. On the contrary, the important message could be shown in a better way. One of the most important designs is the predefined slits. Authors mark it as C1-C4 in Fig.2 c,d,e. However, it is still very difficult for readers to see these slits, especially in a printed version. The same suggestions also apply to the text, some details could be refined or moved to the supplementary information. The key message, i.e. how the predefined slits work, worths a better explanation.

[1] Sequential metamaterials with alternating Poisson's ratios.
<https://doi.org/10.1038/s41467-022-28696-9>

[2] Multi-step self-guided pathways for shape-changing metamaterials.
<https://doi.org/10.1038/s41586-018-0541-0>

[3] Enhanced stiffness characteristic and anisotropic quasi-static compression properties of a negative Poisson's ratio mechanical metamaterial.
<https://doi.org/10.1016/j.tws.2022.109757>

[4] Extended Poisson's Ratio Range in Chiral Isotropic Elastic Material.
<https://doi.org/10.1002/pssb.202200336>

[5] Metamaterials with Poisson's ratio sign toggling by means of microstructural duality.
<https://doi.org/10.1007/s42452-019-0185-1>

Reviewer #2 (Remarks to the Author):

In this work, the authors present a class of 'self-bridging' metamaterials which undergo two different deformation modes upon tensile and compressive loading. In the former case, the deformations are the typical metamaterial modes of rotations and ligament flexure while in the latter, contacts between

'unconnected' components result in different deformations. This has the potential to result in unconventional properties such as non-reciprocity.

Although the article is extremely well-written, I have some major concerns about the level of novelty of this work. The concept of non-reciprocal metamaterials has been around for some time, while the use of internal contacts to induce different deformation modes under compressive loading than those observed under tensile loading is also not new. The metamaterial topologies investigated are also relatively well-known, with Structures A and C being geometrically modified variants of the classic anti-tetrachiral and rotating squares systems respectively. In view of this, I do not believe that this work meets the criteria for novelty for publication in Nature Communications. Other comments:

a) The periodic boundary conditions presented in the Supplementary Information appear to be highly restrictive and, although probably accurate and sufficient for simulating a representative unit cell with quadratic symmetry, may not be sufficient if an axi-asymmetric unit cell such as the one presented in Supplementary Figure 2. Have the authors validated this method with respect to more generalised methods such as the ones presented in the following works?

i. <https://www.sciencedirect.com/science/article/pii/S0020768305001460>

ii. <https://link.springer.com/article/10.1007/s00366-018-0616-4>

iii. <https://link.springer.com/article/10.1007/s00366-019-00910-1>

b) The authors argue throughout the manuscript that these self-bridging metamaterials overcome the classic laws of orthotropy. While in practical terms, this statement may be considered somewhat accurate, the laws of classic elasticity do not take into account the fact that the microstructure of the material is 'different' for compressive and tensile loading, i.e. contacts are active for compression and not for tension. These laws are typically accurate for linear, infinitesimally small strain loading and under these conditions the deformation mechanism should theoretically be the same for both tensile and compressive loading (since at infinitesimally small strains contact of components is not yet activated). I think that it is important to mention this point in the discussion – not to decrease the validity of the authors' claims, but merely to indicate how the limits of classic theoretical elasticity were bypassed rather than broken through this method.

Response to Reviewers

Reviewer #1

The authors introduce three types of self-bridging metamaterials, which achieve non-trivial Poisson's ratios, respectively non-reciprocal Poisson's ratios, ultra-large Poisson's ratios and step-like Poisson's ratios. The three metamaterials are designed based on predefined slits using topology optimization. The results have been verified by experiments, simulations and equivalent models.

The main novelty of this article is presenting the concept of three non-trivial Poisson's ratios using self-bridging metamaterials. Many studies [#1][#2] have used self-contact (or self-bridging) to design mechanical metamaterials. The first property, non-reciprocal Poisson's ratio widely exist in anisotropy materials. It is not surprising that the anisotropy structure the authors presented shows non-reciprocal Poisson's ratio. For example, Fig.3(a) and (b) of the ref[#3] clearly show non-reciprocal Poisson's ratio. The second property, ultra-large Poisson's ratios, has been studied in ref[#4] and also shown in Fig.3(a) of the ref[#3]. The third property, step-like Poisson's ratio also has been studied by several researchers, see Fig.1 of the ref [#5]. I agree that although all concepts have been studied before, to the best of my knowledge, the combination of these concepts is new. But the fundamental rule or the design guide of the presented self-bridging metamaterials is unclear to me. The design is based on the topology optimization, it is unclear that which part of the structure is essential to achieve the properties that the authors presented.

[#1] Sequential metamaterials with alternating Poisson's ratios. <https://doi.org/10.1038/s41467-022-28696-9>

[#2] Multi-step self-guided pathways for shape-changing metamaterials. <https://doi.org/10.1038/s41586-018-0541-0>

[#3] Enhanced stiffness characteristic and anisotropic quasi-static compression properties of a negative Poisson's ratio mechanical metamaterial. <https://doi.org/10.1016/j.tws.2022.109757>

[#4] Extended Poisson's Ratio Range in Chiral Isotropic Elastic Material. <https://doi.org/10.1002/pssb.202200336>

[#5] Metamaterials with Poisson's ratio sign toggling by means of microstructural duality. <https://doi.org/10.1007/s42452-019-0185-1>

Reply:

Thanks for the reviewer's very valuable comments and questions. This article aims to realize metamaterials with non-trivial Poisson's ratios that have prescribed values bypassing the thermodynamic limit ($0 \leq \nu_{ij}\nu_{ji} < 1$). To this end, in this article we developed an inverse design framework. The design guide of the framework resides in the fact that slits are predefined for self-contacts in the initial design region and then the material distribution in the remaining region is determined by topology optimization to pursue target Poisson's ratios. For the inverse design framework, both predefined slits and topology optimization are essential, which is explained in response to **Comment 3** in detail. The essential geometrical parameters for each type of designed metamaterials are explained in response to **Comment 3**. For each type of designed metamaterials, we analyze in detail the deformation mechanism that enables non-trivial Poisson's ratios in response to **Comment 1**.

Within this design framework, we realized non-reciprocal Poisson's ratios ($v_{ij}v_{ji} < 0$) and ultra-large Poisson's ratios ($v_{ij}v_{ji} > 1$), respectively. In addition, the inverse design framework can also produce step-like Poisson's ratios (compressive $v_{ij} \neq$ tensile v_{ji}). The main novelty and contribution of this article are summarized as follows.

Realization of metamaterials with non-reciprocal, ultra-large, and step-like Poisson's ratios, respectively. In this article, each type of non-trivial Poisson's ratios is designed independently by setting the target Poisson's ratios in topology optimization formulation. For the non-reciprocal Poisson's ratios, we unveil unprecedented bulk modes that break static reciprocity. For the ultra-large Poisson's ratios, the designed metamaterial exhibits orthogonally bidirectional displacement amplification, where the output displacement can be more than 3 times the input displacement even though input and output locations are swapped. For the step-like Poisson's ratios, the designed metamaterial can transversely expand under both longitudinal tension and compression.

Inverse design framework based on predefined slits for self-contacts and topology optimization. Contact, geometric, and material nonlinearities are basic nonlinear sources in mechanics. Though it has been demonstrated that designing metamaterial with these nonlinearities can generate exotic properties or functionalities [#1-#2], the introduction of geometric and material nonlinearities in topology optimization has failed so far to produce Poisson's ratios bypassing the thermodynamic limit [#6-#7]. To the best of the authors' knowledge, there was no inverse design framework based on contacts and topology optimization to explore such non-trivial Poisson's ratios. The contribution and advancement of the developed inverse design framework in this article are demonstrated by the realizations of metamaterials with Poisson's ratios bypassing the thermodynamic limit.

Compared with studies [#1-#5], the difference and novelty of this article are discussed in terms of the design framework and realized non-trivial Poisson's ratios as follows.

Fig. #1: Unit cells. a, Unit cell in the study [#1]. b, Unit cell in the study [#2]. c, Unit cell in this article.

Design framework. The unit cell from each design framework of the studies [#1-#2] and this article is presented in Fig. #1. These design frameworks have different target properties for designed metamaterials.

The study [#1] realized sequential metamaterials with alternating Poisson's ratios. The study [#2] proposed shape-changing metamaterials that undergo self-guided and multi-step reconfiguration. However, this article developed an inverse design framework to realize metamaterials with target Poisson's ratios bypassing the thermodynamic limit. Though self-contacts are considered in these design frameworks, they have significantly different distributions of self-contacts and microstructural topologies, which are specially built for the target properties. Thus, compared with studies[#1-#2], this article has a novel design framework in terms of realized non-trivial properties and design ideas.

The studies [#1-#2] were cited in references [6,11] of the revised manuscript, respectively. The comparison has been added to the revised manuscript:

"The distributions of self-contacts induced by predefined slits and the topology of the microstructures are significantly different from those in other design frameworks using self-contacts^{6,11}, which are built for different properties."

Fig. #2: Unit cell in the study [#3] and its simulations. a, d, Microstructure of the unit cell and its simulation in the study [#3], respectively. **b, e**, Unit cell built by a continuum and its simulation via homogenization method, respectively. **c, f**, Structure with 6×6 unit cells and its simulation to validate the homogenization

method, respectively. **g, h, i**, Poisson's ratio ν , elastic modulus E and $E\nu$ calculated by the unit cell (Eff.) and structure (Struc.) under different structural angles θ , respectively.

Non-reciprocal Poisson's ratio. It is worth noting that for isotropic, orthotropic, and anisotropic materials, their constitutive tensors are symmetric, namely $E_j\nu_{ji} = E_i\nu_{ij}$, due to static Maxwell–Betti reciprocity. And, the thermodynamic limit of Poisson's ratio ($0 \leq \nu_{ij}\nu_{ji} < 1$) allows $\nu_{ji} \neq \nu_{ij}$. In this article, the non-reciprocal Poisson's ratios are defined as $\nu_{ij}\nu_{ji} < 0$, instead of $\nu_{ji} \neq \nu_{ij}$. With positive elastic modules, the non-reciprocal Poisson's ratios can result in $E_2\nu_{21} > 0 > E_1\nu_{12}$ as shown in Fig. 3c of the revised manuscript, which breaks the symmetry of constitutive tensors ($E_j\nu_{ji} = E_i\nu_{ij}$) and unveils unprecedented static non-reciprocal modes as shown in Fig. 3d of the revised manuscript. The relationship between non-reciprocal Poisson's ratios $\nu_{12}\nu_{21} < 0$ and $F_1u_{2 \rightarrow 1} \neq F_2u_{1 \rightarrow 2}$ is derived in the supplementary information.

To show the static reciprocity, the metamaterial (Fig. #2a) from the study [#3] is simulated by a continuum as shown in Fig. #2b and e, where the thickness t is the same as the data given in the study [#3]. For different structural angles θ , the Poisson's ratio ν , elastic modulus E , and $E\nu$ are provided in Fig. #2g-i, respectively, which are calculated by the homogenization (representative volume element) method (Fig. #2b and e) with the linear elastic assumption that is the same as the setting in the study [#3] ('When considering the y-direction loading, a small external force along the y direction works on point B to assure the linear elastic deformation.' [#3]). The result of the homogenization method is validated by the simulation of the macroscopic structure (Fig. #2c and f). Though the metamaterial can show different Poisson's ratios (Fig. #2g) and elastic moduli (Fig. #2h) in different directions, $\nu_{12}\nu_{21} > 0$ (Fig. #3) and $E_2\nu_{21} = E_1\nu_{12}$ (Fig. #2i), which can also be demonstrated by the symmetric compliance matrices in Eq. (#1)-(#4). Thus, we believe that the study [#3] did not realize non-reciprocal Poisson's ratios. In this article, we realized $\nu_{12}\nu_{21} < 0$ and $E_2\nu_{21} > 0 > E_1\nu_{12}$ as shown in Fig. 3a and c of the revised manuscript, which unveils unprecedented bulk modes that break static reciprocity as shown in Fig. 3d of the revised manuscript. To the best of the authors' knowledge, there are no studies that explore metamaterials with non-reciprocal Poisson's ratios and unveil the static non-reciprocity therein.

$$\begin{bmatrix} 1.57\text{E}-9 & -1.09\text{E}-10 & 0 \\ -1.09\text{E}-10 & 1.64\text{E}-9 & 0 \\ 0 & 0 & 2.08\text{E}-8 \end{bmatrix} \quad (\#1)$$

$$\begin{bmatrix} 2.00\text{E}-9 & 3.47\text{E}-10 & 0 \\ 3.47\text{E}-10 & 1.20\text{E}-9 & 0 \\ 0 & 0 & 2.48\text{E}-8 \end{bmatrix} \quad (\#2)$$

$$\begin{bmatrix} 2.59\text{E}-9 & 6.02\text{E}-10 & 0 \\ 6.02\text{E}-10 & 8.16\text{E}-10 & 0 \\ 0 & 0 & 2.60\text{E}-8 \end{bmatrix} \quad (\#3)$$

$$\begin{bmatrix} 3.06E-9 & 3.89E-10 & 0 \\ 3.89E-10 & 3.68E-10 & 0 \\ 0 & 0 & 2.64E-8 \end{bmatrix} \quad (\#4)$$

The study [#3] is cited in references [12] of the revised manuscript. The definition of the non-reciprocal Poisson's ratio has been added to the revised manuscript:

"Poisson's ratio, metamaterials have been realized to support negative values based on auxetic deformation patterns, via bendable or buckled ligaments⁸⁻¹², or rotatable nodes¹³⁻¹⁵."

"enabling inaccessible deformation patterns, including one-way displacement amplification with broken reciprocity (non-reciprocal Poisson's ratios, $v_{ij}v_{ji} < 0$, in Fig. 1c and f)"

Fig. #3: $v_{12}v_{21}$ calculated by the unit cell (Eff.) of the study [#3] and structure (Struc.) under different structural angles θ .

Ultra-large Poisson's ratio. The thermodynamic limit of Poisson's ratio in orthotropic material is $0 \leq v_{ij}v_{ji} < 1$. In this article, $v_{ij}v_{ji} > 1$ is called ultra-large Poisson's ratio. $v_{ij}v_{ji}$ of the metamaterial in the study [#3] is calculated as shown in Fig. #3. It can be seen that $v_{ij}v_{ji}$ in the study [#3] was not beyond 1. Thus, we believe that the study [#3] did not realize ultra-large Poisson's ratios. In this article, we realize the metamaterial with $v_{12} \approx v_{21} > 3$ as shown in Fig. 3e of the revised manuscript, which is the evidence of $v_{ij}v_{ji} > 1$ (with a maximum value of approximately 10).

In this article, the thermodynamic limit of Poisson's ratio is derived from classical elasticity based on traditional Hooke's law [#8], which does not describe any chiral effects [#9]. It is also the theoretical basis of the design framework in this article. In addition, the designed microstructures have mirror symmetry and anti-chirality as shown in Fig.2 of the revised manuscript. However, in the study [#4], the range of Poisson's ratio is theoretically analyzed by chiral Cosserat elasticity (also called micropolar elasticity) mechanics, where chirality is a basic assumption for the microstructure [#4]. Compared with the theoretical analysis in the study [#4], this article designed and realized the metamaterial with ultra-large Poisson's ratio via different basic mechanics theories.

The studies [#4, #8, #9] were cited in references [17, 16, 7] of the revised manuscript, respectively. We have explained the thermodynamic limit of Poisson's ratio in the revised manuscript:

"thermodynamics predicts a general bound on the Poisson's ratios in the linear and stable elastic regime¹⁶,

namely $0 \leq \nu_{ij}\nu_{ji} < 1$ (Fig. 1f), where ν_{ij} and ν_{ji} are Poisson's ratios in two orthogonal directions ($i = 1, 2, 3; j = 1, 2, 3; i \neq j$) and micropolar elasticity is not considered¹⁷."

Fig. #4: Design framework in the study [#5]. **a**, Schematic illustration of the design framework. **b**, Unit cell with discrete rigid body.

Step-like Poisson's ratio. The study [#5] proposed a design framework (Fig. #4a) of the discrete rigid body (Fig. #4b) to obtain compressive $\nu_{12} > 0$ and tensile $\nu_{12} < 0$. However, the design framework in this article is built for the continuum. Thus, in terms of step-like Poisson's ratio, the design framework in this article can realize step-like Poisson's ratio in the continuum that is not covered by the study [#5].

In studies [#10, #11], the metamaterials contract under both tension and compression as shown in Fig. #5. Thus, their Poisson's ratios are compressive $\nu_{12} < 0$ and tensile $\nu_{12} > 0$. However, this article realizes metamaterial expands under both tension and compression (compressive $\nu_{12} > 0$ and tensile $\nu_{12} < 0$) as shown in Fig. 3f of the revised manuscript. To the best of the authors' knowledge, the design frameworks in studies [#10, #11] can not realize metamaterials that transversely expand under both longitudinal tension and compression. Thus, in terms of step-like Poisson's ratio, the design framework in this article can realize expansion under both longitudinal tension and compression that can not be obtained by studies [#10, #11].

Fig. #5: Design frameworks in the studies [#10-#11]. **a**, Schematic illustration of the design framework and its deformation patterns in the study [#10]. **b**, Schematic illustration of the design framework and its deformation patterns in the study [#11].

The studies [#5, #10, #11] were cited in references [15, 40, 41] of the revised manuscript, respectively. The comparison has been added to the revised manuscript:

"The metamaterial is designed in the continuum instead of a discrete rigid body¹⁵, and it realizes the expansibility under both tension and compression, different from the contractility induced by bending beams^{40,41}"

[#6] Topology optimized architectures with programmable Poisson's ratio over large deformations.
<https://doi.org/10.1002/adma.201502485>

[#7] Nonlinear homogenization for topology optimization.
<https://doi.org/10.1016/j.mechmat.2020.103324>

[#8] Poisson's ratio in orthotropic materials. <https://doi.org/10.2514/3.4974>

[#9] Three-dimensional mechanical metamaterials with a twist.
<https://www.science.org/doi/abs/10.1126/science.aao4640>

[#10] Composite microstructures with Poisson's ratio sign switching upon stress reversal.
<https://doi.org/10.1016/j.compstruct.2018.10.074>

[#11] On the mechanical properties of centro-symmetric honeycombs with T-shaped joints.
<https://doi.org/10.1002/pssb.201384248>

Here more detailed comments and requests for clarification:

Comment 1: I like that the authors use a lever with two fulcrums (Fig.1 h) to explain how to achieve a bidirectional amplification. However, I did not fully understand how the authors apply this multiple-fulcrum lever mechanism to the three designs in Fig 2. c,d and e. I could see both Fig.1 h and Fig. 2d have fulcrums and both show a 'bidirectional' amplification, but I could not see a further connection.

Reply:

Thanks to the reviewer for the very valuable comments and questions. Like a single-mode lever with one fixed fulcrum (Fig. 1g in the revised manuscript), the thermodynamic limit of Poisson's ratio ($0 \leq \nu_{ij}\nu_{ji} < 1$) only allows displacement amplification in one direction via a Poisson's ratio greater than 1. However, a multi-mode lever with two detachable fulcrums can overcome this limit and realize bidirectional displacement amplification by changing rotation modes via different fulcrums (Fig. 1g in the revised manuscript). In terms of internal connectivity, the lever system is reconfigured due to the transition between the connection and separation states of the fulcrums and the lever, and then the lever shows different rotation modes. Inspired by the potential of topological reconfiguration of detachable fulcrums, in this article, the detachable fulcrums are mimicked by predefined slits for the continuum. In terms of internal connectivity, the transition between the connection (self-bridging) and separation states of the predefined slits reconfigures the microstructural topology. Due to the change in the topological configuration of microstructures, constitutive tensors of the microstructure may no longer obey the symmetry and invariance. We can therefore achieve metamaterials with non-trivial Poisson's ratios. With predefined slits, an inverse design framework is developed to design

metamaterials with target Poisson's ratios bypassing the thermodynamic limit in this article. Through the design framework, this article not only realizes ultra-large Poisson's ratios that make designed metamaterial amplify displacement bidirectionally, but also non-reciprocal and step-like Poisson's ratios.

Thus, the multi-mode lever with two detachable fulcrums in Fig. 1g of the revised manuscript is a special case that inspires the predefined slits in our design framework. Like detachable fulcrums in the multi-mode lever, predefined slits in the designed metamaterials enable the topological reconfiguration of a microstructure under different load cases, making the microstructure exhibit different rotation behaviors. For each type of Poisson's ratios, the working mechanism of predefined slits is intuitively analyzed and explicitly validated by the homogenization of equivalent models in the revised manuscript.

Non-reciprocal Poisson's ratios. In the numerical simulations as shown in Fig. 2c of the revised manuscript, " all microstructural slits open under transverse compression. However, when the microstructure is longitudinally compressed, the self-bridging of slits at c_2 and c_4 reconfigures the microstructural topology with a change in internal connectivity. Then, the microscale levers show two different rotation modes: (1) for transverse compression, the microstructure rotates about two fulcrums; (2) for longitudinal compression, the arms at c_2 and c_4 drive the rotation of the microstructure. These two different rotation behaviors result in positive and negative Poisson's ratios, $\nu_{21} > 0$ and $\nu_{12} < 0$, respectively (Fig. 3a)."

Two equivalent models (Supplementary Fig. 4a and b) are built for the metamaterial under two load cases. "The evaluated effective compliance matrices of our equivalent model are significantly different (Supplementary equations S.(13) and (14)), demonstrating topological reconfiguration of the metamaterial induced by self-bridging slits. Thus, the self-bridging feature of the metamaterial enables the variation of constitutive tensors under different load scenarios, which breaks the symmetry $E_j \nu_{ji} = E_i \nu_{ij}$ in static reciprocity and enables bypassing the lower bound ($\nu_{ij} \nu_{ji} \geq 0$)."

Ultra-large Poisson's ratios. In the numerical simulations as shown in Fig. 2d of the revised manuscript, "the locations of rotation fulcrums (self-contacts) under transverse compression are c_1 and c_3 , while the locations change to c_2 and c_4 under longitudinal compression (Fig. 2d). In terms of internal connectivity, different locations of self-contacts make the microstructure exhibit different topological configuration. The length of microscale levers between two fulcrums increases under compression, which amplifies the input displacement, and then the changed rotation fulcrums lead to amplification in both orthogonal directions. The bidirectional displacement amplification mechanism of the metamaterial is similar to a multi-mode lever with two detachable fulcrums, where input displacements on both the left and right can be amplified by changing the fulcrum (Fig. 1g)."

Two equivalent models (Supplementary Fig. 4c and d) are built for the metamaterial under two load cases. "The upper bound $\nu_{ij} \nu_{ji} < 1$ is deduced by assuming an invariant positive-definite matrix to ensure a positive strain energy density. However, in the effective compliance matrices of these two equivalent models (Supplementary equations S.(15) and (16)), the values of the first and second elements on the principal diagonal are swapped, which demonstrates that the microstructural topology of the metamaterial is reconfigured as the locations of the self-bridging slits change. The constitutive tensors of the designed

metamaterial vary under different load cases, and thus the upper bound can be violated by the self-bridging metamaterial."

Step-like Poisson's ratios. In the numerical simulations as shown in Fig. 2e of the revised manuscript, under compression, "in terms of internal connectivity, the microstructural topology is reconfigured by self-bridging slits at c_2 and c_4 in the microstructure (Fig. 2e). Then, the microscale levers show different rotation modes: (1) Under tension, all slits are pulled apart, and the microstructure rotates with anti-chirality; (2) Under compression, c_1 and c_3 open while c_2 and c_4 close, and the microstructure rotates about fulcrums at c_2 and c_4 . Only the first rotation mode activates the auxeticity, and thus the metamaterial exhibits step-like Poisson's ratios."

Two equivalent models (Supplementary Fig. 4e and f) are built for the metamaterial under two load cases. "The calculated effective compliance matrices of these two equivalent models are significantly different (Supplementary equations S.(17) and (18)). Because the constitutive tensors of the designed metamaterial are no longer invariant under different loads, the self-bridging metamaterial can show different Poisson's ratios under different loads."

Comment 2: Authors design the three metamaterials using topology optimization. The structure shown in Fig. 2c shows non-reciprocal Poisson's ratios, but it should also be able to show the step-like Poisson's ratios, i.e. compressive $\nu_{12} < 0$ and tensile $\nu_{12} > 0$. Why do the authors design another structure shown in Fig. 2e? Similarly, if there are predefined slits at position C1 and C3 as well in the structure shown in Fig. 2c, then it should also be able to achieve ultra-large Poisson's ratio, i.e. compressive $\nu_{21} > 1$, compressive $\nu_{12} > 1$. So by only adjusting the predefined slits, all three non-trivial Poisson's ratios can be realized based on the orange structure (Fig. 2c). But the authors design three different structures to show three non-trivial properties of Poisson's ratios. Do different designs add any new information? If it is just due to topology optimization, the novelty and influence will be greatly reduced.

Reply:

Thanks for the reviewer's very valuable questions. The aim of this article is to realize metamaterials with non-trivial Poisson's ratios that have prescribed values bypassing the thermodynamic limit. To this end, this article developed an inverse design framework based on predefined slits for self-contacts and topology optimization. In the framework, the design problem is described by an optimization formulation in Eq. (7), where the target Poisson's ratios are realized by constraining the error between actual and prescribed values. Then, each type of non-trivial Poisson's ratios is designed independently under prescribed corresponding Poisson's ratios in the optimization formulation. Though, in post-evaluation, the designed metamaterial with non-reciprocal Poisson's ratios can show step-like without postprocessing and $\nu_{ij}\nu_{ji} > 1$ with postprocessing, each type of designed metamaterial has its specialty, which is discussed as follows.

Comparison between designed metamaterials with non-reciprocal and step-like Poisson's ratios. The designed metamaterial with non-reciprocal Poisson's ratios is obtained with target Poisson's ratios $\nu_{12}^* = -0.5$

and $\nu_{21}^* = 1.4$. In addition to non-reciprocity, it is able to transversely contract under both longitudinal tension and compression, i.e., compressive $\nu_{12} < 0$ and tensile $\nu_{12} > 0$. The designed metamaterial with step-like Poisson's ratios is obtained with target compressive $\nu_{12}^* = 0.8$ and tensile $\nu_{12}^* = -0.8$. It transversely expands under both longitudinal tension and compression, i.e., compressive $\nu_{12} > 0$ and tensile $\nu_{12} < 0$.

Comparison between designed metamaterials with non-reciprocal and ultra-large Poisson's ratios. If more slits at c_1 and c_3 are added into the designed metamaterial with non-reciprocal Poisson's ratios, then it is able to achieve compressive $\nu_{12} > 1$ and compressive $\nu_{21} > 1$. After postprocessing, the metamaterial with non-reciprocal Poisson's ratios has C_4 rotation symmetry. Thus, compressive ν_{12} is equal to compressive ν_{21} . Based on Fig. 3a, after postprocessing, the product $\nu_{ij}\nu_{ji}$ is about 2. The designed metamaterial with ultra-large Poisson's ratios is obtained with target $\nu_{21}^* = \nu_{12}^* = 3$. It can reach large $\nu_{ij}\nu_{ji}$, i.e., $\nu_{ij}\nu_{ji} \approx 10$, as shown in Fig. 3e, which is an order of magnitude larger than the limit $\nu_{ij}\nu_{ji} < 1$.

Comment 3: So the main question would be what is the design guide? Which geometry parameters are essential to achieve the non-trivial Poisson's ratios? By adding predefined slits, the symmetry of tension and compression is broken, so ν_{12} (compressive) $\neq \nu_{12}$ (tensile). So all the three structures should show 'step-like Poisson's ratios'. The green one and blue one have step-like Poisson's ratio along both horizontal and vertical directions. The orange one has a step-like Poisson's ratio only along vertical direction. The green one and blue one also have C_4 rotation symmetry, which makes $\nu_{12} = \nu_{21}$. Therefore, if $\nu_{12} > 1$, then $\nu_{21} > 1$ as well, named 'ultra-large Poisson's ratios' in the article. While the orange one does not have C_4 rotation symmetry, so $\nu_{12} \neq \nu_{21}$, named non-reciprocal Poisson's ratios in the article. On the top of the symmetries, if the chiral motion is triggered, then the Poisson's ratio is positive.

Reply:

Thanks for the reviewer's very valuable questions. To realize metamaterials with non-trivial Poisson's ratios that have prescribed values bypassing the thermodynamic limit, this article developed an inverse design framework. The design guide of the framework is that slits are predefined for self-contacts in the initial design region and then the material distribution in the remaining region is determined by topology optimization to pursue target Poisson's ratios. For the inverse design framework, both predefined slits and topology optimization are essential to achieve the non-trivial Poisson's ratios, which are explained as follows.

Predefined slits. Like a single-mode lever with one fixed fulcrum (Fig. 1g), the thermodynamic limit ($0 \leq \nu_{ij}\nu_{ji} < 1$) only allows displacement amplification in one direction via a Poisson's ratio greater than 1. However, a multi-mode lever with two detachable fulcrums can overcome this limit and realize bidirectional displacement amplification by changing rotation modes via different fulcrums (Fig. 1g). In terms of internal connectivity, the lever system is reconfigured due to the transition between the connection and separation states of the fulcrums and the lever, and then the lever shows different rotation modes. Inspired by the potential of topological reconfiguration of detachable fulcrums, in this work, the detachable fulcrums are mimicked by predefined slits for the continuum. In terms of internal connectivity, the transition between the connection

(self-bridging) and separation states of the predefined slits reconfigures the microstructural topology. Due to the change of the topological configuration of microstructures, constitutive tensors of the microstructure may no longer obey the symmetry and invariance. We can therefore achieve the metamaterials with Poisson's ratios bypassing the thermodynamic limit.

Topology optimization. When there is insufficient prior knowledge of microstructures, such as some regular geometries, realizing metamaterials with prescribed properties is a challenge for trial-and-error and parameter optimization. In terms of design flexibility, topology optimization that presents optimal material distribution for microstructures can provide much higher design freedom than trial-and-error and parameter optimization. In this article, slits are predefined for self-contacts in the initial design region, and the remaining design region is discretized into about 10000 design elements. The independent design region has been illustrated in Supplementary Fig. 1.

Fig. #6: Design framework in this article. **a**, Microstructural symmetry and locations of the predefined slits. The independently designed region is indicated in green, and the orange 'F' is used to indicate the symmetry in the designed region. **b**, Enlarged view of independent design region and fixed solid region for the predefined slits. The unit cell is discretized into about 10000 design elements. The width w of the slits is $0.0002a$. $L_3 = 0.01a - 0.5w$. For the non-reciprocal Poisson's ratio, $L_1 = 0.15a$, $L_2 = 0.1a$. For the ultra-large and step-like Poisson's ratios, $L_1 = 0.1a$, $L_2 = 0.05a$. **c**, Microstructural topology of the metamaterial with step-like Poisson's ratio in the independent design region obtained by the inverse design framework of this article. **d**, Microstructural topology of the metamaterial with step-like Poisson's ratios. **e**, 2×2 designed unit cells of the metamaterial with step-like Poisson's ratios. **f**, Mirror symmetry in the metamaterial. The locations of the

predefined slits are on the axis of the mirror symmetry.

In the inverse design framework, in terms of the geometry parameters and optimization formulation, the essential differences in the realization of each type of the non-trivial Poisson's ratios are discussed as follows.

Geometry parameters. The independently designed region in the approach is indicated in green as shown in Fig. #6a, and orange 'F' is used to indicate the symmetry in the cell. The cell has four-fold rotational symmetry in the entire domain and mirror symmetry exists in its quarter at each corner as shown in Fig. #6a. The slits are distributed along the axis of mirror symmetry as shown in Fig. #6f. For the design region, the essential difference in the realization of each type of the non-trivial Poisson's ratios is the setting of predefined slits. Four slits (c_1 , c_2 , c_3 , and c_4) are predefined for the design of ultra-large and step-like Poisson's ratios, while only two slits (c_2 and c_4) are predefined for the design of non-reciprocal Poisson's ratios. For topology optimization, the geometrical boundaries are determined by the elemental design variables. Hence, topology optimization produces freeform geometry, instead of regular geometry determined by some parameters.

Optimization formulation. In the design framework, the design problem is described by an optimization formulation in Eq. (7) of the revised manuscript, where the target Poisson's ratios are realized by constraining the error between actual and prescribed values. The design framework aims to independently realize each type of non-trivial Poisson's ratios and pursues prescribed values of Poisson's ratios, instead of one metamaterial with all features of three types of non-trivial Poisson's ratios. In Fig. 2c, d, and e of the revised manuscript, the designed unit cells with non-reciprocal, ultra-large, and step-like Poisson's ratios are obtained with target Poisson's ratios $\nu_{21}^* = 1.4$ and $\nu_{12}^* = -0.5$, $\nu_{21}^* = \nu_{12}^* = 3$, and tensile $\nu_{12}^* = -0.8$ and compressive $\nu_{12}^* = 0.8$, respectively. For three types of metamaterials, the update processes of the microstructural topologies are presented in Fig. #7.

Fig. #7: Update processes of microstructural topologies for three types of metamaterials. a, Non-

reciprocal Poisson's ratios. **b**, Ultra-large Poisson's ratios. **c**, Step-like Poisson's ratios. The number of iterations (Iter.) is listed under the microstructures.

The differences in the realization of each type of the non-trivial Poisson's ratios are explained in the 'Design, simulations, and experiments' section of the revised manuscript. The setting of geometrical parameters has been added to Supplementary Fig. 1. The update processes of the microstructural topologies have been added to Supplementary Fig. 8.

"The slits are distributed along the axis of mirror symmetry. Two slits (c_2 and c_4) are predefined for the design of non-reciprocal Poisson's ratios, while four slits (c_1 , c_2 , c_3 , and c_4) are predefined for the design of ultra-large and step-like Poisson's ratios. The geometrical parameters of the slits are provided in Supplementary Fig. 1."

Though the designed metamaterials in post-evaluation are able to exhibit multiple types of non-trivial Poisson's ratios, each one has its specialty, which is difficult to reach for others. The specialty of each designed metamaterial is discussed as follows.

Non-reciprocal Poisson's ratios. It is worth noting that the non-reciprocal Poisson's ratios are $v_{ij}v_{ji} < 0$, instead of $v_{ji} \neq v_{ij}$. The designed metamaterial with non-reciprocal Poisson's ratios exhibits $E_2v_{21} > 0 > E_1v_{12}$ (Fig. 3c in the revised manuscript), which demonstrates that the designed metamaterial breaks the symmetry of constitutive tensors ($E_jv_{ji} = E_iv_{ij}$) in static reciprocity and predicts the extreme static non-reciprocity $F_1u_{2 \rightarrow 1} > 0 > F_2u_{1 \rightarrow 2}$. The unprecedented bulk mode that breaks static reciprocity is validated in Fig. 4d and Fig. 3f.

Ultra-large Poisson's ratios. The designed metamaterial with ultra-large Poisson's ratios is obtained with target $v_{21}^* = v_{12}^* = 3$. It can bidirectionally amplify input displacement three times and reach large $v_{ij}v_{ji}$, i.e., $v_{ij}v_{ji} \approx 10$ as shown in Fig. 3e, which is an order of magnitude larger than the limit $v_{ij}v_{ji} < 1$. The large $v_{ij}v_{ji}$ is difficult to achieve for other designed metamaterials with postprocessing.

Step-like Poisson's ratios. The designed metamaterial with step-like Poisson's ratios is obtained with target compressive $v_{12}^* = 0.8$ and tensile $v_{12}^* = -0.8$. It is able to transversely expands under both longitudinal tension and compression. As shown in Fig. 3f, in the simulation, the compressive $v_{12} \approx 0.8$ and tensile $v_{12} \approx -0.8$. This large expansibility in both tension and compression is difficult to reach for other designed metamaterials with postprocessing.

Comment 4: There is still some space to improve the structure of the manuscript. Some parts are redundant. For example, Fig.1 a-e show different combinations of poisson's ratios, but Fig. 1f summarizes all in a concise way. Fig 1 i,j,k and the top row of Fig 2 c,d,e show the same design. Similarly, Fig.2 c,d,e and Fig 4 show the information as well. Fig.2 f and g doesn't add much information to this article. On the contrary, the important message could be shown in a better way. One of the most important designs is the predefined slits. Authors mark it as C1-C4 in Fig.2 c,d,e. However, it is still very difficult for readers to see these slits, especially in a

printed version. The same suggestions also apply to the text, some details could be refined or moved to the supplementary information. The key message, i.e. how the predefined slits work, worths a better explanation.

Reply:

Thanks for the reviewer's very important comments and suggestions. We have simplified the figures in the revised manuscript. Fig. 1i, j, and k and Fig. 2 f and g in the original manuscript were deleted. In order to intuitively and explicitly compare each type of Poisson's ratios, Fig. 1a-f in the original manuscript is retained in the revised manuscript. To validate the deformation patterns in numerical simulation, Fig. 4 in the original manuscript is retained in the revised manuscript.

To clearly show the predefined slits in the designed metamaterials, their local views have been added to Supplementary Fig. 1. The text has been refined, and the details of the method have been moved to the supplementary information.

In the 'Non-reciprocal Poisson's ratios', 'Ultra-large Poisson's ratios', and 'Step-like Poisson's ratios' sections of the revised manuscript, the working mechanisms are explained in detail, respectively, including intuitive analysis and explicit validation based on homogenization of equivalent models.

Reviewer #2

In this work, the authors present a class of ‘self-bridging’ metamaterials which undergo two different deformation modes upon tensile and compressive loading. In the former case, the deformations are the typical metamaterial modes of rotations and ligament flexure while in the latter, contacts between ‘unconnected’ components result in different deformations. This has the potential to result in unconventional properties such as non-reciprocity.

Comment 1: Although the article is extremely well-written, I have some major concerns about the level of novelty of this work. The concept of non-reciprocal metamaterials has been around for some time, while the use of internal contacts to induce different deformation modes under compressive loading than those observed under tensile loading is also not new. The metamaterial topologies investigated are also relatively well-known, with Structures A and C being geometrically modified variants of the classic anti-tetrachiral and rotating squares systems respectively. In view of this, I do not believe that this work meets the criteria for novelty for publication in Nature Communications.

Reply:

Thanks for the reviewer’s very important comments. This article aims to realize metamaterials with Poisson’s ratios bypassing the thermodynamic limit ($0 \leq v_{ij}v_{ji} < 1$). To the best of the authors’ knowledge, there is no design framework to explore and realize such metamaterials. To this end, this article developed an inverse design framework, realizing non-reciprocal Poisson’s ratios ($v_{ij}v_{ji} < 0$) and ultra-large Poisson’s ratios ($v_{ij}v_{ji} > 1$). In addition, the inverse design framework can produce step-like Poisson’s ratios (compressive $v_{ij} \neq$ tensile v_{ij}). The main novelty and contribution of this article are discussed as follows.

Non-reciprocal Poisson’s ratios. Though the concept of non-reciprocal metamaterials has been around for some time, existing studies mainly break the dynamic (non-zero frequency) reciprocity that the frequency-response functions between any two material points remain the same after swapping source and receiver. This article breaks the static (zero frequency) reciprocity [#12]. In the study [#12], the basic idea to realize static non-reciprocity ($F_{AuB \rightarrow A} \neq F_{BuA \rightarrow B}$) is breaking geometric symmetry and introducing geometric nonlinearity as shown in Fig. #8a. In this article, we open a novel avenue to realize static non-reciprocity, i.e., inverse design of the metamaterial with Poisson’s ratios $v_{ij}v_{ji} < 0$ as shown in Fig. #8b. The designed metamaterial breaks the symmetry of constitutive tensors ($E_j v_{ji} = E_i v_{ij}$) and unveils unprecedented bulk modes of static non-reciprocity, as shown in Fig. #9c-e. Thus, the Poisson’s ratios $v_{ij}v_{ji} < 0$ are called non-reciprocal Poisson’s ratios. Poisson’s ratio is a fundamental property that directly quantifies the relation between input (u_i) and output ($u_{i \rightarrow j}$) displacement. Hence, our designed metamaterial may provide an explicit and programmable way to manipulate the non-reciprocal transmission of the displacement field via designing Poisson’s ratios.

In this article, the relationship between non-reciprocal Poisson’s ratios $v_{12}v_{21} < 0$ and $F_1 u_{2 \rightarrow 1} \neq F_2 u_{1 \rightarrow 2}$ is derived as follows (The relationship is demonstrated in the Supplementary Information).

Fig. #8 Static non-reciprocal modes in the study [#12] and this article. **a**, Metamaterial and static non-reciprocal modes induced by geometric nonlinearity and broken geometric symmetry [#12]. **b**, Metamaterial and static non-reciprocal modes in this article realized by the inverse design of Poisson's ratios bypassing the thermodynamic limit, i.e., $v_{ij}v_{ji} < 0$.

The compliance matrix \mathbf{C} in the orthotropic linear elastic regime is

$$\mathbf{C} = \begin{bmatrix} 1/E_1 & -v_{12}/E_2 & 0 \\ -v_{21}/E_1 & 1/E_2 & 0 \\ 0 & 0 & 1/G \end{bmatrix} \quad (\#1)$$

where E_1 and E_2 are the elastic moduli in the x_1 and x_2 direction, respectively. G is the shear modulus. The symmetry of the compliance matrix gives

$$E_1v_{12} = E_2v_{21} \quad (\#2)$$

The elastic modulus (E_1 or E_2) can be calculated using the stress (F_1/A or F_2/A) and strain (u_1/a or u_2/a), where F_1 (F_2) is the reaction force of the unit cell with periodic boundary conditions under the strain u_1/a (u_2/a). u_1 and u_2 are displacements in the x_1 and x_2 directions, respectively. a is lattice constant. $A = at$, where t is the thickness of the unit cell. Then, $E_1v_{12} = E_2v_{21}$ can be rewritten as

$$\frac{F_1a}{u_1A}v_{12} = \frac{F_2a}{u_2A}v_{21} \quad (\#3)$$

$$F_1v_{12}u_2 = F_2v_{21}u_1 \quad (\#4)$$

Based on $v_{12} = -u_1/u_2$ and $v_{21} = -u_2/u_1$, the input displacement u_2 (u_1) in the x_2 (x_1) direction outputs

the displacement $u_{2 \rightarrow 1}$ ($u_{1 \rightarrow 2}$) in the x_1 (x_2) direction. The relation between Poisson's ratios and these displacements is

$$-v_{12}u_2 = u_{2 \rightarrow 1} \quad (\#5)$$

$$-v_{21}u_1 = u_{1 \rightarrow 2} \quad (\#6)$$

Then, equation (#4) can be expressed as a reciprocal formulation:

$$F_1u_{2 \rightarrow 1} = F_2u_{1 \rightarrow 2} \quad (\#7)$$

In the linear elastic regime, for a general continuum with positive elastic moduli E_1 and E_2 , both v_{12} and v_{21} are simultaneously positive, zero, or negative; then, $0 \leq v_{ij}v_{ji}$. Based on equations (#2) and (#7), bypassing this limit to make $v_{21} > 0$ and $v_{12} < 0$ can offer an unusual deformation pattern, i.e., non-reciprocal transmission of displacement fields, where $E_2v_{21} > 0 > E_1v_{12}$ and $F_1u_{2 \rightarrow 1} > 0 > F_2u_{1 \rightarrow 2}$. In this article, we realize the metamaterial with non-reciprocal Poisson's ratios ($v_{12}v_{21} < 0$) as shown in Fig. #9a that break the symmetry of constitutive tensors, i.e. $E_2v_{21} > 0 > E_1v_{12}$ (Fig. #9c), unveiling unprecedented bulk modes that break static reciprocity, i.e. $F_1u_{2 \rightarrow 1} > 0 > F_2u_{1 \rightarrow 2}$ (Fig. #9d).

Fig. #9: Poisson's ratio, elastic moduli, and non-reciprocity evaluations. **a**, Non-reciprocal Poisson's ratios. "FEA," "Exp.," "Eqv.," and "Eff." correspond to simulations of designed unit cells, experiments on designed unit cells, simulations of equivalent models, and effective material parameters, respectively. **b**, Elastic moduli E_1 and E_2 of the metamaterial with non-reciprocal Poisson's ratios in two orthogonal directions. **c**, Products of elastic moduli and Poisson's ratios. **d**, Output displacement $u_{1 \rightarrow 2}$ ($u_{2 \rightarrow 1}$) and input displacement u_1 (u_2) for different values of F_1 (F_2). **e**, Experiments about non-reciprocity under transverse and longitudinal loads. The structure is supported by rollers that are compressed using an aluminum alloy plate.

Ultra-large Poisson's ratios. Mathematically, the compliance matrix has to be positive-definite to ensure a positive strain energy density [#13]. Then, the Poisson's ratios in the linear orthotropic constitutive law are thermodynamically constrained by $\nu_{ij}\nu_{ji} < 1$. Hence, Poisson's ratios larger than 1 in both orthogonal directions and the deformation pattern of orthogonally bidirectional displacement amplification are inaccessible in the linear elastic regime. So far, the introduction of geometric and material nonlinearities has failed to produce Poisson's ratios bypassing the thermodynamic limit [#14-#15]. However, in this article, we developed an inverse design framework based on predefined slits for self-contacts and topology optimization, realizing the metamaterial with ultra-large Poisson's ratios, i.e., $\nu_{ij}\nu_{ji} > 1$, as shown in Fig. #10d. The designed metamaterial can bidirectionally amplify input displacement three times and reach large $\nu_{ij}\nu_{ji}$, i.e., $\nu_{ij}\nu_{ji} \approx 10$, as shown in Fig. #10, which is an order of magnitude larger than the limit ($\nu_{ij}\nu_{ji} < 1$), as shown in Fig. #10b-d.

Fig. #10: Designed metamaterial with ultra-large Poisson's ratios and its deformation patterns. **a**, Designed microstructure including 2×2 unit cells with lattice constant a , one of which is boxed by red dashed lines. Prescribed slits are labeled c_1, c_2, c_3 , and c_4 . **b, c**, Deformation patterns. Each color bar shows the output displacement field in the x_1 or x_2 direction. The rotation modes and fulcrums are denoted by arrows and balls, respectively. **d**, Ultra-large Poisson's ratios.

Step-like Poisson's ratios. Under a small strain in the linear elastic regime, the constitutive relation is mathematically invariant. Consequently, the Poisson's ratio in the linear elastic regime does not change either under tensile or compressive strain in a particular direction, which is demonstrated by a line with a slope of 1 in Fig. 1f of the revised manuscript. Hence, ordinary materials in the linear elastic regime cannot achieve step-like Poisson's ratios that are negative and positive under longitudinal tension and compression, respectively, and cannot exhibit the deformation pattern of transverse expansion under both load cases (Fig. 1e of the revised manuscript). Some studies based on discrete rigid bodies [#16] and bending beams [#17, #18] explored the step-like Poisson's ratios. Different from these works, in this article, we developed a new inverse design framework of continuum based on predefined slits for self-contacts and topology optimization to pursue prescribed step-like Poisson's ratios. The designed metamaterial with step-like Poisson's ratios (Fig. #8a) exhibits transverse expansion under both longitudinal tension and compression (Fig. #11b-c). As shown in

Fig. #11d, in the simulation, the compressive $\nu_{12} \approx 0.8$ and tensile $\nu_{12} \approx -0.8$, which demonstrates the step-like Poisson's ratio and the large expansibility under both tension and compression. In terms of realized metamaterials with step-like Poisson's ratios, we have explained the difference between the studies [#16-#18] and this article: "The metamaterial is designed in the continuum instead of a discrete rigid body¹⁵, and it realizes the expansibility under both tension and compression, different from the contractility induced by bending beams^{40,41}."

Fig. #11: Designed metamaterial with step-like Poisson's ratios and its deformation patterns. a, Designed microstructure including 2×2 unit cells. **b, c,** Deformation patterns. The rotation modes and fulcrums are denoted by arrows and balls, respectively. **d,** Step-like Poisson's ratios.

Inverse design framework. Though it has been demonstrated that designing metamaterial with geometric, material, or contact nonlinearities can generate exotic properties or functionalities, the introduction of geometric and material nonlinearities has failed so far to produce Poisson's ratios bypassing the thermodynamic limit [#14, #15]. To the best of the authors' knowledge, there was no inverse design framework with contacts to explore and realize the non-trivial Poisson's ratios bypassing the thermodynamic limit.

Like a single-mode lever with one fixed fulcrum (Fig. 1g in the revised manuscript), the thermodynamic limit ($0 \leq \nu_{ij}\nu_{ji} < 1$) only allows displacement amplification in one direction via a Poisson's ratio greater than 1. However, a multi-mode lever with two detachable fulcrums can overcome this limit and realize bidirectional displacement amplification by changing rotation modes via different fulcrums (Fig. 1g in the revised manuscript). Inspired by the detachable fulcrums, in this article, the detachable fulcrums are mimicked by predefined slits for the continuum. Then, to realize metamaterials with target Poisson's ratios bypassing the thermodynamic limit, this article develops an inverse design framework based on predefined slits for self-contact and topology optimization, as shown in Fig. #12.

Fig. #12: Design framework in this article. **a**, Microstructural symmetry and locations of the predefined slits. The independently designed region is indicated in green, and the orange ‘F’ is used to indicate the symmetry in the designed region. **b**, Enlarged view of independent design region and fixed solid region for the predefined slits. The unit cell is discretized into about 10000 design elements. The width w of the slits is $0.0002a$. $L_3 = 0.01a - 0.5w$. For the non-reciprocal Poisson’s ratios, $L_1 = 0.15a$, $L_2 = 0.1a$. For the ultra-large and step-like Poisson’s ratios, $L_1 = 0.1a$, $L_2 = 0.05a$. **c**, Microstructural topology of the metamaterial with step-like Poisson’s ratios in the independent design region obtained by the inverse design framework of this article. **d**, Microstructural topology of the metamaterial with step-like Poisson’s ratios. **e**, 2×2 designed unit cells of the metamaterial with step-like Poisson’s ratios. **f**, Mirror symmetry in the metamaterial. The locations of the predefined slits are on the axis of the mirror symmetry.

The novelty of topologies of designed metamaterials. Metamaterials with positive or negative Poisson’s ratios have been realized in some microstructural configurations, such as the anti-tetrachiral and rotating squares systems. However, to the best of the authors’ knowledge, there is no microstructural configuration to realize metamaterials with Poisson’s ratios bypassing the thermodynamic limit. In this case, it is difficult to realize Poisson’s ratios bypassing the thermodynamic limit via modifying geometrical parameters of the classic anti-tetrachiral and rotating squares, because there is no sufficient prior knowledge about microstructural configurations that provides guides for modification.

In this article, these metamaterials with Poisson’s ratios bypassing the thermodynamic limit are obtained by the developed inverse design framework with topology optimization, instead of trial-and-error based on

the classic microstructural configurations. In terms of design flexibility, topology optimization that presents optimal material distribution for microstructures can provide much higher design freedom than trial-and-error and parameter optimization. In this article, slits are predefined for self-contacts in the initial design region, and the remaining design region is discretized into about 10000 design elements. For three types of metamaterials, the update processes of the microstructural topologies are presented in Fig. #13. It can be observed that microstructural topologies change dramatically from initial guesses to the final designs after topology optimization, which is the evidence that our designed metamaterials are not obtained by simply modifying the geometrical parameters of the classic anti-tetrachiral and rotating squares. The update processes of the microstructural topologies are added to Supplementary Fig. 8.

Fig. #13: Update processes of microstructural topologies for three types of metamaterials. a, Non-reciprocal Poisson's ratios. **b,** Ultra-large Poisson's ratios. **c,** Step-like Poisson's ratios. The number of iterations (Iter.) is listed under the microstructures.

[#12] Static non-reciprocity in mechanical metamaterials. <https://doi.org/10.1038/nature21044>

[#13] Poisson's ratio in orthotropic materials. <https://doi.org/10.2514/3.4974>

[#14] Topology optimized architectures with programmable Poisson's ratio over large deformations. <https://doi.org/10.1002/adma.201502485>

[#15] Nonlinear homogenization for topology optimization. <https://doi.org/10.1016/j.mechmat.2020.103324>

[#16] Metamaterials with Poisson's ratio sign toggling by means of microstructural duality. <https://doi.org/10.1007/s42452-019-0185-1>

[#17] Composite microstructures with Poisson's ratio sign switching upon stress reversal. <https://doi.org/10.1016/j.compstruct.2018.10.074>

[#18] On the mechanical properties of centro-symmetric honeycombs with T-shaped joints.

Comment 2: The periodic boundary conditions presented in the Supplementary Information appear to be highly restrictive and, although probably accurate and sufficient for simulating a representative unit cell with quadratic symmetry, may not be sufficient if an axi-asymmetric unit cell such as the one presented in Supplementary Figure 2. Have the authors validated this method with respect to more generalised methods such as the ones presented in the following works?

[#19] <https://www.sciencedirect.com/science/article/pii/S0020768305001460>

[#20] <https://link.springer.com/article/10.1007/s00366-018-0616-4>

[#21] <https://link.springer.com/article/10.1007/s00366-019-00910-1>

Reply:

Thanks for the reviewer’s careful observation and valuable questions. We validated the periodic boundary conditions (PBCs) used in this article via comparison with the PBCs given in the study [#19]. The PBCs in the study [#19] are realized in the Cell Periodicity node in the Solid Mechanics interface in COMSOL Multiphysics. The result of the comparison is presented in Fig. #14. It can be seen that the deformations and Poisson’s ratios obtained based on the two types of PBCs are almost the same. It demonstrates the correctness of the PBCs used in this article.

Fig. #14: Comparisons between the periodic boundary conditions (PBCs) used in this article and given

in the study [#19] based on the designed metamaterials. **a.** Undeformed unit cells. **b.** Deformed unit cells under longitudinal compression with PBCs given in the study [#19]. **c.** Deformed unit cells under longitudinal compression with PBCs in this article.

Because the designed metamaterials have mirror symmetry as shown in Fig. #15, their constitutive relations are considered orthotropy and the shear strain between top and bottom boundaries is zero, i.e. $u_1(K) = u_1(K')$ as shown in Fig. #14b. Thus, for the designed metamaterials, $u_{12} = u_{14}$ of PBCs used in this article does not restrict more degree of freedom.

For the description of the PBCs in Supplementary Fig. 2 of the original Supplementary Information, we are so sorry that the orthotropy required by the PBCs used in this article is neglected. In the revised Supplementary Fig. 2, we have selected an example with mirror symmetry as shown in Fig. #16 to describe the PBCs used in this article. The studies [#19-#21] were cited in references [36-38] of the revised manuscript. And, in the revised manuscript, we have added the explanation: "In this work, periodic boundary conditions are constructed based on orthotropy. More general periodic boundary conditions can be found in the studies³⁶⁻³⁸."

Fig. #15: Microstructures with 2×2 designed unit cells of different types of Poisson’s ratios. a. Non-reciprocal Poisson’s ratio. **b.** Ultra-large Poisson’s ratio. **c.** Step-like Poisson’s ratio.

Fig. #16: Periodic boundary conditions. a. Metamaterial with periodically arranged unit cells. **b.** Periodic boundary conditions: $u_{13} - u_{11} = u_1, u_{21} = u_{23}, u_{24} - u_{22} = u_2, u_{12} = u_{14}$.

Comment 3: The authors argue throughout the manuscript that these self-bridging metamaterials overcome

the classic laws of orthotropicity. While in practical terms, this statement may be considered somewhat accurate, the laws of classic elasticity do not take into account the fact that the microstructure of the material is 'different' for compressive and tensile loading, i.e. contacts are active for compression and not for tension. These laws are typically accurate for linear, infinitesimally small strain loading and under these conditions the deformation mechanism should theoretically be the same for both tensile and compressive loading (since at infinitesimally small strains contact of components is not yet activated). I think that it is important to mention this point in the discussion – not to decrease the validity of the authors' claims, but merely to indicate how the limits of classic theoretical elasticity were bypassed rather than broken through this method.

Reply:

Thanks for the reviewer's very valuable comments and suggestions. The thermodynamic limit of Poisson's ratios ($0 \leq \nu_{ij}\nu_{ji} < 1$) is derived from the linear elastic regime, where infinitesimally small strain is a basic assumption. This point is mentioned in the manuscript: "The thermodynamic limit is derived from the linear elastic regime with the assumption of infinitesimally small strain, yet the introduction of nonlinearities has so far failed to produce Poisson's ratios bypassing the limit^{27,28}".

This article realizes metamaterials with Poisson's ratios bypassing the thermodynamic limit, which is emphasized in the Discussion of the revised manuscript.

"Mechanical metamaterials bypassing the thermodynamic limit of Poisson's ratios were designed and realized through inverse design based on predefined slits for self-contacts and topology optimization. Symmetry and invariance of constitutive tensors are basic assumptions for the theoretical limit of Poisson's ratios in the linear elastic regime. In terms of internal connectivity, the microstructural topologies of the designed mechanical metamaterials are reconfigured by the self-bridging slits, driving different rotation behaviors of microscale levers. Then, under different loads, the constitutive tensors of the self-bridging metamaterials no longer obey symmetry and invariance, enabling Poisson's ratios that bypass the conventional limits."

REVIEWERS' COMMENTS

Reviewer #1 (Remarks to the Author):

I have read the revised manuscript carefully. The authors have answered most of the questions in detail. However, the authors have not made many major revisions. The doubts I mentioned last time about the novelty of the paper have not been allayed. Therefore, I am reluctant to recommend the present manuscript for publication.

I respect the authors' design principles, slits + topology, to achieve non-trivial Poisson's ratios bypassing the thermodynamic limit. However, as the authors agree with, one structure presented by the authors can achieve both non-reciprocal and step-like Poisson's ratios. And the authors design two different topologies to achieve non-reciprocal and step-like Poisson's ratios, respectively. From this point of view, I personally think that the design principle presented by the authors is not general enough. Without a general design principle, the novelty of the paper may not be enough to be published in Nature Communications, as these three types of non-trivial Poisson's ratios have already been studied by other researchers.

Reviewer #2 (Remarks to the Author):

The authors have addressed all my concerns and have clearly demonstrated the novelty of their work. I recommend acceptance.

Response to Reviewers

Reviewer #1

I have read the revised manuscript carefully. The authors have answered most of the questions in detail. However, the authors have not made many major revisions. The doubts I mentioned last time about the novelty of the paper have not been allayed. Therefore, I am reluctant to recommend the present manuscript for publication.

I respect the authors' design principles, slits + topology, to achieve non-trivial Poisson's ratios bypassing the thermodynamic limit. However, as the authors agree with, one structure presented by the authors can achieve both non-reciprocal and step-like Poisson's ratios. And the authors design two different topologies to achieve non-reciprocal and step-like Poisson's ratios, respectively. From this point of view, I personally think that the design principle presented by the authors is not general enough. Without a general design principle, the novelty of the paper may not be enough to be published in Nature Communications, as these three types of non-trivial Poisson's ratios have already been studied by other researchers.

Reply:

Thanks for the reviewer's very valuable comments. This article aims to realize metamaterials with non-trivial Poisson's ratios that have prescribed values surpassing the thermodynamic limit ($0 \leq v_{ij}v_{ji} < 1$). To this end, we developed an inverse design framework of predefined slits and topology optimization. It is worth noting that we aim to realize each type of non-trivial Poisson's ratios **independently**, instead of one metamaterial with all these types of non-trivial Poisson's ratios. We have emphasized this point in the section of Design, simulations, and experiments: "the topological configuration of each mechanical metamaterial is independently designed by setting target Poisson's ratios in the optimization formulation".

The inverse design framework can realize non-reciprocal Poisson's ratios ($v_{ij}v_{ji} < 0$), ultra-large Poisson's ratios ($v_{ij}v_{ji} > 1$), and step-like Poisson's ratios (compressive $v_{ij} \neq$ tensile v_{ji}). For the non-reciprocal Poisson's ratios, we unveil an unprecedented bulk mode that breaks static reciprocity. For the ultra-large Poisson's ratios, the designed metamaterial exhibits orthogonally bidirectional displacement amplification, where the output displacement can be more than 3 times the input displacement even though input and output locations are swapped. For the step-like Poisson's ratios, the designed metamaterial can transversely expand under both longitudinal tension and compression. **The inverse design framework is general** because each type of non-trivial Poisson's ratios can be designed by setting the target Poisson's ratios in topology optimization formulation.

In terms of the novelty of realized Poisson's ratios, we have carefully compared with our work and studies [#1-#9] in the response to your comments last time:

Non-reciprocal Poisson's ratio. It is worth noting that for isotropic, orthotropic, and anisotropic materials, their constitutive tensors are symmetric, namely $E_{jv_{ji}} = E_{iv_{ij}}$, due to static Maxwell-Betti reciprocity. And, the thermodynamic limit of Poisson's ratio ($0 \leq v_{ij}v_{ji} < 1$) allows $v_{ji} \neq v_{ij}$. **In this article, the non-reciprocal Poisson's ratios are defined as $v_{ij}v_{ji} < 0$, instead of $v_{ji} \neq v_{ij}$.** With positive elastic modules, the non-reciprocal Poisson's ratios can result in $E_{2v_{21}} > 0 > E_{1v_{12}}$ as shown in Fig. 3c of the revised manuscript, which breaks the

symmetry of constitutive tensors ($E_j\nu_{ji} = E_i\nu_{ij}$) and unveils unprecedented static non-reciprocal modes as shown in Fig. 3d of the revised manuscript. The relationship between non-reciprocal Poisson's ratios $\nu_{12}\nu_{21} < 0$ and $F_1u_{2\rightarrow 1} \neq F_2u_{1\rightarrow 2}$ is derived in the supplementary information.

Fig. #1: Unit cell in the study [#1] and its simulations. **a, d**, Microstructure of the unit cell and its simulation in the study [#1], respectively. **b, e**, Unit cell built by a continuum and its simulation via homogenization method, respectively. **c, f**, Structure with 6×6 unit cells and its simulation to validate the homogenization method, respectively. **g, h, i**, Poisson's ratio ν , elastic modulus E and $E\nu$ calculated by the unit cell (Eff.) and structure (Struc.) under different structural angles θ , respectively.

To show the static reciprocity, the metamaterial (Fig. #1a) from the study [#1] is simulated by a continuum as shown in Fig. #1b and e, where the thickness t is the same as the data given in the study [#1]. For different structural angles θ , the Poisson's ratio ν , elastic modulus E , and $E\nu$ are provided in Fig. #1g-i, respectively, which are calculated by the homogenization (representative volume element) method (Fig. #1b and e) with the linear elastic assumption that is the same as the setting in the study [#1] ('When considering the y-direction loading, a small external force along the y direction works on point B to assure the linear elastic deformation.' [#1]). The result of the homogenization method is validated by the simulation of the macroscopic structure (Fig. #1c and f). Though the metamaterial can show different Poisson's ratios (Fig. #1g) and elastic moduli (Fig. #1h) in different directions, $\nu_{12}\nu_{21} > 0$ (Fig. #2) and $E_2\nu_{21} = E_1\nu_{12}$ (Fig. #1i), which can also be demonstrated by the symmetric compliance matrices in Eq. (#1)-(#2). **Thus, we believe that the study [#1]**

did not realize non-reciprocal Poisson's ratios. In this article, we realized $\nu_{12}\nu_{21} < 0$ and $E_2\nu_{21} > 0 > E_1\nu_{12}$ as shown in Fig. 3a and c of the revised manuscript, which unveils unprecedented bulk modes that break static reciprocity as shown in Fig. 3d of the revised manuscript. To the best of the authors' knowledge, there are no studies that explore metamaterials with non-reciprocal Poisson's ratios and unveil the static non-reciprocity therein.

$$\begin{bmatrix} 1.57\text{E}-9 & -1.09\text{E}-10 & 0 \\ -1.09\text{E}-10 & 1.64\text{E}-9 & 0 \\ 0 & 0 & 2.08\text{E}-8 \end{bmatrix} \quad (\#1)$$

$$\begin{bmatrix} 2.00\text{E}-9 & 3.47\text{E}-10 & 0 \\ 3.47\text{E}-10 & 1.20\text{E}-9 & 0 \\ 0 & 0 & 2.48\text{E}-8 \end{bmatrix} \quad (\#2)$$

$$\begin{bmatrix} 2.59\text{E}-9 & 6.02\text{E}-10 & 0 \\ 6.02\text{E}-10 & 8.16\text{E}-10 & 0 \\ 0 & 0 & 2.60\text{E}-8 \end{bmatrix} \quad (\#3)$$

$$\begin{bmatrix} 3.06\text{E}-9 & 3.89\text{E}-10 & 0 \\ 3.89\text{E}-10 & 3.68\text{E}-10 & 0 \\ 0 & 0 & 2.64\text{E}-8 \end{bmatrix} \quad (\#4)$$

The study [#1] is cited in references [12] of the revised manuscript. The definition of the non-reciprocal Poisson's ratio has been added to the revised manuscript:

"Poisson's ratio, metamaterials have been realized to support negative values based on auxetic deformation patterns, via bendable or buckled ligaments⁸⁻¹², or rotatable nodes¹³⁻¹⁵."

"enabling inaccessible deformation patterns, including one-way displacement amplification with broken reciprocity (non-reciprocal Poisson's ratios, $\nu_{ij}\nu_{ji} < 0$, in Fig. 1c and f)"

Fig. #2: $\nu_{12}\nu_{21}$ calculated by the unit cell (Eff.) of the study [#1] and structure (Struc.) under different structural angles θ .

Ultra-large Poisson's ratio. The thermodynamic limit of Poisson's ratio in orthotropic material is $0 \leq \nu_{ij}\nu_{ji} < 1$. In this article, $\nu_{ij}\nu_{ji} > 1$ is called ultra-large Poisson's ratio. $\nu_{ij}\nu_{ji}$ of the metamaterial in the study [#1] is calculated as shown in Fig. #2. It can be seen that $\nu_{ij}\nu_{ji}$ in the study [#1] was not beyond 1. **Thus, we**

believe that the study [#1] did not realize ultra-large Poisson's ratios. In this article, we realize the metamaterial with $\nu_{12} \approx \nu_{21} > 3$ as shown in Fig. 3e of the revised manuscript, which is the evidence of $\nu_{ij}\nu_{ji} > 1$ (with a maximum value of approximately 10).

In this article, the thermodynamic limit of Poisson's ratio is derived from classical elasticity based on traditional Hooke's law [#6], which does not describe any chiral effects [#7]. It is also the theoretical basis of the design framework in this article. In addition, the designed microstructures have mirror symmetry and anti-chirality as shown in Fig.2 of the revised manuscript. However, in the study [#2], the range of Poisson's ratio is theoretically analyzed by chiral Cosserat elasticity (also called micropolar elasticity) mechanics, where chirality is a basic assumption for the microstructure [#2]. Compared with the theoretical analysis in the study [#2], this article designed and realized the metamaterial with ultra-large Poisson's ratio via different basic mechanics theories.

The studies [#2, #6, #7] were cited in references [17, 16, 7] of the revised manuscript, respectively. We have explained the thermodynamic limit of Poisson's ratio in the revised manuscript:

"thermodynamics predicts a general bound on the Poisson's ratios in the linear and stable elastic regime¹⁶, namely $0 \leq \nu_{ij}\nu_{ji} < 1$ (Fig. 1f), where ν_{ij} and ν_{ji} are Poisson's ratios in two orthogonal directions ($i = 1, 2, 3; j = 1, 2, 3; i \neq j$) and micropolar elasticity is not considered¹⁷."

Step-like Poisson's ratio. The study [#3] proposed a design framework (Fig. #3a) of the discrete rigid body (Fig. #3b) to obtain compressive $\nu_{12} > 0$ and tensile $\nu_{12} < 0$. However, the design framework in this article is built for the continuum. **Thus, in terms of step-like Poisson's ratio, the design framework in this article can realize step-like Poisson's ratio in the continuum that is not covered by the study [#3].**

Fig. #3: Design framework in the study [#3]. a, Schematic illustration of the design framework. **b,** Unit cell with discrete rigid body.

In studies [#8, #9], the metamaterials contract under both tension and compression as shown in Fig. #4. Thus, their Poisson's ratios are compressive $\nu_{12} < 0$ and tensile $\nu_{12} > 0$. However, this article realizes metamaterial expands under both tension and compression (compressive $\nu_{12} > 0$ and tensile $\nu_{12} < 0$) as shown in Fig. 3f of the revised manuscript. To the best of the authors' knowledge, the design frameworks in studies [#8, #9] can not realize metamaterials that transversely expand under both longitudinal tension and compression. **Thus, in terms of step-like Poisson's ratio, the design framework in this article can realize**

expansion under both longitudinal tension and compression that can not be obtained by studies [#8, #9].

Fig. #4: Design frameworks in the studies [#8, #9]. **a**, Schematic illustration of the design framework and its deformation patterns in the study [#8]. **b**, Schematic illustration of the design framework and its deformation patterns in the study [#9].

The studies [#3, #8, #9] were cited in references [15, 40, 41] of the revised manuscript, respectively. The comparison has been added to the revised manuscript:

"The metamaterial is designed in the continuum instead of a discrete rigid body¹⁵, and it realizes the expansibility under both tension and compression, different from the contractility induced by bending beams^{40,41}"

- [#1] Enhanced stiffness characteristic and anisotropic quasi-static compression properties of a negative Poisson's ratio mechanical metamaterial. <https://doi.org/10.1016/j.tws.2022.109757>
- [#2] Extended Poisson's Ratio Range in Chiral Isotropic Elastic Material. <https://doi.org/10.1002/pssb.202200336>
- [#3] Metamaterials with Poisson's ratio sign toggling by means of microstructural duality. <https://doi.org/10.1007/s42452-019-0185-1>
- [#4] Topology optimized architectures with programmable Poisson's ratio over large deformations. <https://doi.org/10.1002/adma.201502485>
- [#5] Nonlinear homogenization for topology optimization. <https://doi.org/10.1016/j.mechmat.2020.103324>
- [#6] Poisson's ratio in orthotropic materials. <https://doi.org/10.2514/3.4974>
- [#7] Three-dimensional mechanical metamaterials with a twist. <https://www.science.org/doi/abs/10.1126/science.aao4640>
- [#8] Composite microstructures with Poisson's ratio sign switching upon stress reversal. <https://doi.org/10.1016/j.compstruct.2018.10.074>
- [#9] On the mechanical properties of centro-symmetric honeycombs with T-shaped joints. <https://doi.org/10.1002/pssb.201384248>

Reviewer #2

The authors have addressed all my concerns and have clearly demonstrated the novelty of their work. I recommend acceptance.

Reply:

Thanks for the reviewer's very valuable comments, suggestions and recommendations.